# Principles of RNA recruitment to viral ribonucleoprotein condensates in a segmented dsRNA virus

**Sebastian Strauss[1], Julia Acker[2], Guido Papa[3†], Daniel Desirò[2], Florian Schueder[1,4], Alexander Borodavka[2]\*, Ralf Jungmann[1,4]\***

[1]Max Planck Institute of Biochemistry, Munich, Germany; [2]Department of Biochemistry, University of Cambridge, Cambridge, United Kingdom; [3]Molecular Immunology Laboratory, International Centre for Genetic Engineering and Biotechnology, Trieste, Italy; [4]Department of Physics and Center for Nanoscience, Ludwig Maximilian University, Munich, Germany

**\*For correspondence:**
ab2677@cam.ac.uk (AB);
jungmann@biochem.mpg.de (RJ)

**Present address:** [†]Medical Research Council Laboratory of Molecular Biology (MRC LMB), Cambridge Biomedical Campus, Cambridge, United Kingdom

**Competing interest:** The authors declare that no competing interests exist.

**Abstract** Rotaviruses transcribe 11 distinct RNAs that must be co-packaged prior to their replication to make an infectious virion. During infection, nontranslating rotavirus transcripts accumulate in cytoplasmic protein-RNA granules known as viroplasms that support segmented genome assembly and replication via a poorly understood mechanism. Here, we analysed the RV transcriptome by combining DNA-barcoded smFISH of rotavirus-infected cells. Rotavirus RNA stoichiometry in viroplasms appears to be distinct from the cytoplasmic transcript distribution, with the largest transcript being the most enriched in viroplasms, suggesting a selective RNA enrichment mechanism. While all 11 types of transcripts accumulate in viroplasms, their stoichiometry significantly varied between individual viroplasms. Accumulation of transcripts requires the presence of 3' untranslated terminal regions and viroplasmic localisation of the viral polymerase VP1, consistent with the observed lack of polyadenylated transcripts in viroplasms. Our observations reveal similarities between viroplasms and other cytoplasmic RNP granules and identify viroplasmic proteins as drivers of viral RNA assembly during viroplasm formation.

## Editor's evaluation

Viral replication in the cell requires the assembly of multiple viral components into individual viral particles, while maintaining a relatively strict ratio between individual components. This manuscript uses an imaging approach to study proposed aggregates of viral protein and nucleic acid components referred to as 'viroplasms' and their role in achieving ordered viral assembly. Although this work provides a glimpse of viral assembly in the cell, further work is required to better understand this complex and important process.

## Introduction

RNA genome segmentation poses challenges for assembly and genome packaging of viruses, including rotaviruses, a large group of human and animal pathogens. The rotavirus (RV) genome is enclosed in a protein shell, inside which 11 double-stranded (ds)RNAs, also known as genomic segments, iteratively undergo rounds of transcription (*Caddy et al., 2021*). Consequently, multiple copies of distinct RV transcripts accumulate in the cytoplasm of infected cells *Lu et al., 2008*; *McClain et al., 2010*; *Estrozi et al., 2013*; *Periz et al., 2013*; *Salgado et al., 2017*; *Borodavka et al., 2018*. It remains a long-standing mystery how RVs robustly select and co-package 11 non-identical RNAs

despite the non-stoichiometric transcript accumulation in cells *Ayala-Breton et al., 2009*. While recent multiplexed single-molecule RNA Fluorescence In Situ Hybridisation (smFISH) analyses have identified multi-RNA complexes in influenza A virus-infected cells *Haralampiev et al., 2020*, so far in RVs, the formation of multi-RNA complexes associated with the viral RNA chaperone NSP2 has only been documented in vitro *Borodavka et al., 2017*. Within 2 hr of transcription, viral RNA-binding proteins NSP2 and NSP5 *Patton and Spencer, 2000*; *Taraporewala and Patton, 2004*; *Trask et al., 2012* begin to form membraneless cytoplasmic replication factories, also known as viroplasms *Jayaram et al., 2002*; *Patton et al., 2006*; *Papa et al., 2021*, that accumulate additional viral proteins and RNAs required for subsequent genome replication and assembly. Experimental evidence suggest that viroplasms may provide a selective environment that may protect viral transcripts from siRNA-mediated degradation *Silvestri et al., 2004* where RV transcripts serve as templates for the synthesis of dsRNA genome. Previous attempts to investigate the ultrastructure of viroplasms have not succeeded in revealing the identities and stoichiometry of individual transcripts therein *Patton et al., 2006*; *Crawford and Desselberger, 2016*; *Criglar et al., 2018*; *Garcés Suárez et al., 2019*.

Recently, we discovered that early replication stage (2–6 hr post infection, hpi) viroplasms represent RNA-protein condensates that are formed via phase-separation of the non-structural phosphoprotein NSP5 *Geiger et al., 2021* and the RNA chaperone NSP2. These condensates could be rapidly and reversibly dissolved by treating RV-infected cells with small aliphatic diols *Geiger et al., 2021*. RV transcripts can be released from viroplasms when they were briefly treated with 4.7% propylene glycol or 4% 1,6-hexanediol, followed by reversible recruitment of the RV transcripts into these condensates when these compounds were removed *Geiger et al., 2021*. Paradoxically, while the RNA chaperone NSP2 possesses low nanomolar affinity for any single-stranded (ss)RNA, viroplasms appear to be highly enriched only in viral transcripts *Geiger et al., 2021*. Thus, several aspects of viroplasm formation resemble the assembly of other ribonucleoprotein (RNP) assemblies formed from non-translating mRNAs, for example, stress granules (SGs) *Khong et al., 2017* and P-bodies. For SGs, it has been proposed that essentially every mRNA could partition into these granules, albeit with partitioning efficiencies significantly varying. This suggests that SGs do not represent a defined mRNP assembly but instead form via condensation of non-translating mRNAs and associated proteins in proportion to the RNA length *Khong et al., 2017*. Similarly, efficient partitioning of mRNAs into P-bodies was shown to primarily correlate with their poor levels of translation *Hubstenberger et al., 2017*; *Matheny et al., 2019*. Recent evidence argues that intermolecular RNA-RNA interactions play a role in forming and determining the composition of certain RNP granules *Van Treeck and Parker, 2018*. In principle, the formation of specialised viral ribonucleoprotein condensates could facilitate selective enrichment of untranslated RV transcripts required for a stoichiometric genome assembly via inter-molecular RNA-RNA interactions *Borodavka et al., 2017*; *Bravo et al., 2018*. Despite the extensive evidence of the importance of viroplasms in RV replication *Silvestri et al., 2004*; *Eichwald et al., 2004*; *Taraporewala et al., 2006*; *Vascotto et al., 2004*; *Eichwald et al., 2018*; *Papa et al., 2020*, the analysis of their molecular composition have been confounded by both their dynamic and liquid-like nature that precluded successful isolation from the RV-infected cells. Thus, the exact RNA composition of these assemblies has remained enigmatic, and it is unclear whether these granules contain all 11 non-identical RNA species, and if so, how these organelles maintain their unique RNA composition.

To unravel the principles of RNA partitioning into viroplasms, we have visualised the RV transcriptome using a DNA barcode-based multiplexing approach *Schueder et al., 2017*, combined with single-molecule RNA Fluorescence In Situ Hybridisation (smFISH). Initially, rotavirus transcripts are detected as cytoplasmically distributed non-stoichiometric species, prior to the formation of RNA clusters that precede the assembly of viroplasms. Furthermore, smFISH analysis of individual viroplasms reveals that all RV transcripts are enriched, and that intact 3'UTRs and the viral RNA-dependent RNA polymerase VP1 are required for efficient transcript partitioning into viroplasms. Overall, our data reveal key differences in the mechanisms of RNA partitioning that underlie the assembly of viroplasms and other cytoplasmic RNP granules, including stress granules and P-bodies *Van Treeck and Parker, 2018*; *Wheeler et al., 2016*; *Standart and Weil, 2018*. We propose that VP1-bound viral transcripts undergo viroplasmic enrichment to facilitate segmented RNA genome assortment and assembly.

## Results

### NSP5-EGFP-tagged viral condensates retain viral transcripts

To investigate RV transcript accumulation in viroplasms, we took advantage of the MA104 cell line stably expressing an EGFP-tagged NSP5 that readily distributes into viroplasms *Geiger et al., 2021*; *Eichwald et al., 2004*; *Papa et al., 2019*. At a multiplicity of infection (MOI) of 10, NSP5-EGFP-tagged viroplasms were detected as soon as 2–3 hours post infection (hpi) (*Figure 1*). Recently, we have shown that during these early infection stages, such NSP5-rich granules exhibit liquid-like behaviour, representing dynamic NSP5:NSP2 condensates *Geiger et al., 2021*. To confirm that these EGFP-NSP5-tagged condensates represent bona fide viroplasms that accumulate RV RNA, we used a pooled set of FISH probes consisting of three oligonucleotides targeting protein-coding sequences of each segment of the Rotavirus A genome (G6P6[1] strain RF, further details of probes and fluorophores – see *Supplementary file 1*). Multiple RNA-rich foci were detected in cells (*Figure 1*) as early as 3 hpi, colocalizing with NSP5-EGFP-tagged viroplasms (*Figure 1e*).

Given that each RV transcript is targeted by three transcript-specific probes, and the observed point sources could only be detected after 2–3 hpi, these signals are unlikely to originate from the hybridisation events to single transcripts *Femino et al., 1998*; *Raj et al., 2008*. All newly formed NSP5-EGFP-tagged viroplasms contained RV transcripts (*Figure 1e*), consistent with the notion that these granules represent sites of RV replication *Patton et al., 2006*. We noted that a small fraction of viral transcripts was also detected outside the NSP5-EGFP-tagged granules (*Figure 1*). However, no RV-specific RNA signal was detected in the RV-infected cells up to 3 hpi (*Figure 1*), confirming the specificity of the designed 3 x probes towards the RV transcripts.

Given that EGFP-tagged viroplasms accumulate large amounts of viral RNA-binding proteins *Taraporewala et al., 1999*; *Schuck et al., 2001* known to promiscuously bind non-viral RNAs *Bravo et al., 2018*; *Taraporewala et al., 1999*, we then explored whether other non-viral, highly expressed transcripts, for example, GAPDH, would undergo enrichment in these granules. We performed smFISH to visualise GAPDH transcripts (*Figure 1—figure supplement 1*). The apparent intensity distribution for GAPDH signals was unimodal, as expected for single non-interacting transcripts. While all RV RNA foci (yellow) colocalised with EGFP-tagged viroplasms at 4 hpi (*Figure 1—figure supplement 1*), GAPDH transcripts (red) did not accumulate in viroplasms, suggesting an RNA selection mechanism that determines transcript partitioning into these granules.

We then carried out oligo(dT) FISH *Khong et al., 2017* to visualise sites of accumulation of polyadenylated mRNAs other than GAPDH transcripts. At 4 hpi, RV transcripts partitioned independently of polyadenylated mRNAs (*Figure 1—figure supplement 2*) further confirming that viroplasms are primarily enriched in RV transcripts.

To discern gene-specific viral transcripts, we designed two distinct sets of single-molecule (sm) FISH probes, each targeting the coding regions of the RV gene segments (Seg) Seg3 and Seg4 transcripts, respectively. Since each target-specific pool consisted of 48 probes (**Methods**), at 2 hpi, both Seg3 and Seg4 transcripts were readily detectable as high-intensity single point sources that were sufficiently far apart to be resolved (*Figure 2*). The observed uniformity of intensities of point sources (*Figure 2—figure supplement 1*, **panel a**) was comparable to the GAPDH RNA signal distribution visualised using a set of smFISH probes labelled with an identical fluorophore under the same imaging conditions (**Methods**), further confirming that these objects represented single viral transcripts. Both Seg3 and Seg4 transcripts were equally abundant and randomly distributed in the cytoplasm of infected cells without a discernible pattern. Such random point distribution further suggested a lack of directional transport of RNAs *Femino et al., 1998* released by the transcribing viral particles. The overall cytoplasmic density of transcripts increased over time, reflecting ongoing viral transcription (*Figure 2* and *Figure 2—figure supplement 1*). A hallmark of the rotavirus replication cycle is an exponential increase in the amount of RNA produced after 4–6 hpi *Ayala-Breton et al., 2009*; *Patton et al., 2004* emanating from the second wave of transcription by the newly assembled particles. We therefore initially focused on analysing the intracellular distribution of viral transcripts between 2 and 3 hpi. Despite the apparently equal ratio of Seg3 and Seg4 transcripts produced between 2 and 4 hpi (*Figure 2—figure supplement 1*), at 6 hpi the amount of Seg3 was significantly higher than that of Seg4 RNA (*Figure 2—figure supplement 1*), suggesting that individual viral transcripts have different half-lives. Moreover, after 3 hpi, multiple Seg3 and Seg4 transcripts co-localised, resulting in higher intensity signals compared to single transcripts (*Figure 2* and *Figure 2—figure supplement 1*). The

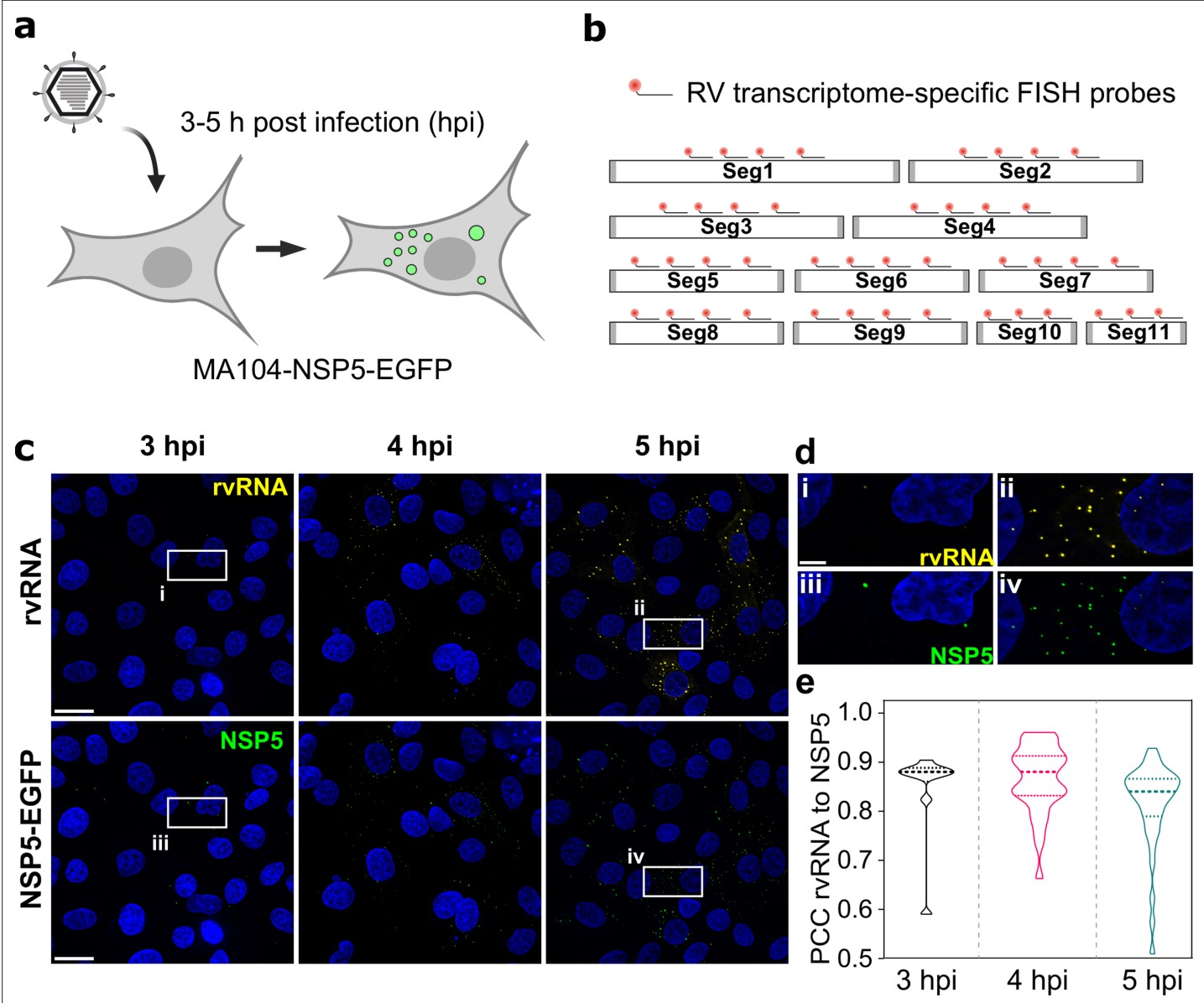

**Figure 1.** RNA ccumulation during Rotavirus infection. (**a**) Experimental design for detecting EGFP-tagged viroplasms during early infection (3–5 hpi) stages. (**b**) Schematics of the FISH probe design for detecting all 11 RV transcripts. A pooled set of 3–4 segment-specific probes labelled with Quasar570 dye were used for visualising RV transcripts in EGFP-tagged viroplasms. (**c**) RV RNA accumulation in viral factories during early infection. Top: RV transcripts (yellow) visualized by smFISH using a combined set of 3–5 segment-specific probes targeting each genomic transcript. Bottom: NSP5-EGFP-tagged viral factories (green). Images represent maximum intensity Z-projections acquired using identical settings and brightness levels for both channels. DAPI-stained nuclei are shown in blue. Scale bars, 25 μm. (**d**) RV transcripts and NSP5-EGFP-specific signals shown in the white box in panel (**c**). Note the low intensity of the RNA and EGFP signals during early infection stages that increase by 5 hpi. Scale bar, 5 μm. (**e**) Pearson's correlation coefficients (PCCs) of colocalising NSP5-EGFP-tagged viral factories and RV transcripts, as shown above in panel (**d**). Dashed and dotted lines represent median and quartile values of PCCs, respectively. Each data point represents the PCC value calculated for a single cell. N=10 (3 hpi), N=31 (4 hpi), N=38 (5 hpi).

The online version of this article includes the following source data and figure supplement(s) for figure 1:

**Source data 1.** Pearson's correlation coefficients (PCCs) of colocalising NSP5-EGFP-tagged viral factories and RV transcripts.

**Source data 2.** Signal intensity distribution histograms for GAPDH transcripts and RV transcript foci.

**Figure supplement 1.** smFISH of GAPDH mRNA and RV transcripts.

**Figure supplement 2.** Viroplasms accumulate RV transcripts but not polyadenylated mRNAs.

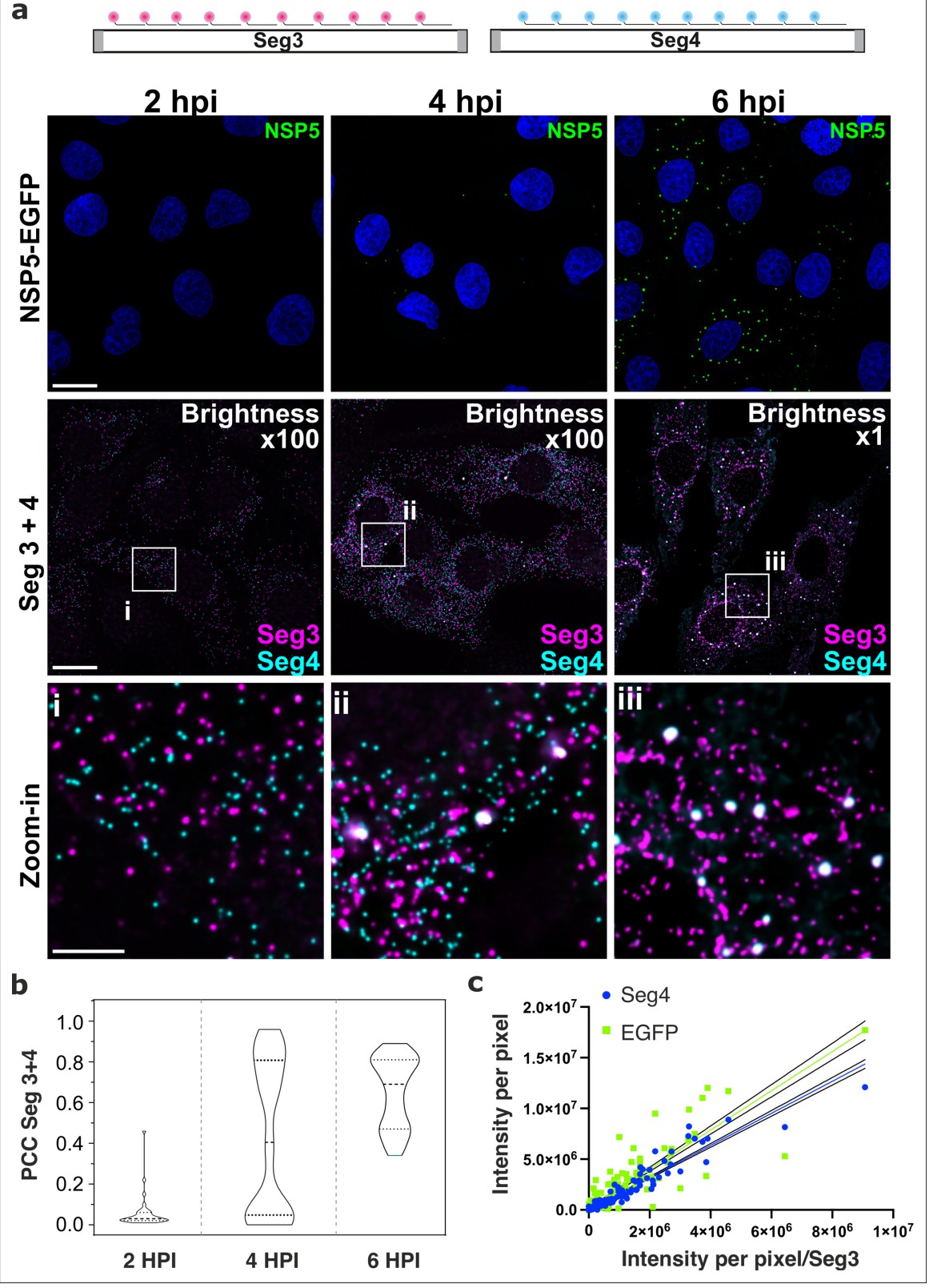

**Figure 2.** Segment-specific transcript accumulation during Rotavirus infection. (**a**) smFISH of RV transcripts at early infection time points (2–6 hpi) using segment-specific probes. Two sets of 48 gene-specific probes were designed for each Seg3 (magenta, Quasar570) and Seg4 (cyan, Quasar670) transcripts. Top: NSP5-EGFP-tagged viral factories (green), nuclei (blue). Middle: Seg3 (magenta) and Seg4 (cyan) RNA signals in the ROIs shown above, with colocalising Seg3 and Seg4 RNAs (white). As the amount of each transcript increases during the course of infection, brightness settings were

*Figure 2 continued on next page*

*Figure 2 continued*

adjusted accordingly to reveal single transcripts (Brightness x100 for 2–4 hpi and 1 x for 6 hpi). Scale bars, 25 μm and 5 μm (zoom-in). (**b**) Colocalisation of Seg3 and Seg4 transcripts. Pearson's correlation coefficients (PCCs) of colocalising Seg3 and Seg4 transcripts. Dashed and dotted lines represent median and quartile values of PCCs, respectively. Each data point represents the PCC value calculated for a single cell. N=37 (2 hpi), N=62 (4 hpi), N=51 (6 hpi). Statistical analysis of data was performed using Kolmogorov-Smirnov test. At the 0.01 level, the observed distributions are significantly different. (**c**) Correlation between Seg4/EGFP and Seg3 RNA signals in RV-infected cells at 4 hpi. Integrated intensities per detected spot for Seg4 RNA, NSP5-EGFP and Seg3 RNAs are plotted to reveal the linear correlation between the signal intensities for each RNA examined, as the intensity of NSP5-EGFP signal increases. Linear regression lines (solid) with 95% CI (dotted lines) are shown for Seg4 RNA (cyan, $R^2$=0.93) and EGFP (green, $R^2$=0.8) intensity signals.

The online version of this article includes the following source data and figure supplement(s) for figure 2:

**Source data 1.** Colocalisation of Seg3 and Seg4 transcripts.

**Source data 2.** Seg4/EGFP and Seg3 RNA signal intensities in RV-infected cells at 4 HPI.

**Source data 3.** Raw signal intensity distributions (counts) of diffraction-limited smFISH-detected spots were derived for Seg3 (Quasar 670 dye-labelled probes) and Seg4 (Quasar 570 dye-labelled probes) transcripts at 2 and 4 hpi.

**Source data 4.** Integrated signal intensities of Seg3, Seg4 transcripts and NSP5-EGFP-tagged viral condensates at different infection time points.

**Source data 5.** Pearson's correlation coefficients (PCCs) calculated for a single cell.

**Source data 6.** Seg4/EGFP and Seg3 RNA signal intensities in RV-infected cells at 6 hpi.

**Source data 7.** Integrated intensities for Seg4 RNA and Seg3 RNA for shNSP2 and WT MA104 cells.

**Source data 8.** Western blot of NSP2 from RV-infected cells (WT MA104 and NSP2-specific shRNA-expressing MA104 cells).

**Figure supplement 1.** Analysis of Seg3 and Seg4 transcript accumulation and localisation in RV-infected cells.

**Figure supplement 2.** NSP2 knockdown in MA104 cells results in the apparent loss of RV RNA clustering.

number of high-intensity RNA foci further increased between 3 and 6 hpi, manifesting in a higher density of co-localising Seg3 and Seg4 transcripts (*Figure 2*). Further analysis of Seg3 and Seg4 RNA intensities of individual viroplasms reveals significant variations in the RNA content between granules. By quantifying Seg3 and Seg4 RNA signals in viroplasms at 4 hpi and 6 hpi (*Figure 2—figure supplement 1d*), a strong correlation ($R^2$=0.93) between Seg4 and Seg3 RNA intensities was noted, with both RNA intensities linearly increase as the intensity of the NSP5-EGFP signal increases, i.e., proportionally to the size of an EGFP-NSP5-tagged viroplasm. Interestingly, a similar Seg3/Seg4 RNA ratio was maintained in viroplasms at a later infection stage at 6 hpi ($R^2$=0.9), while the transcript to NSP5-EGFP ratio was significantly different from that seen at 4 hpi ($R^2$=0.8 and $R^2$ = 0.13 for 4 and 6 hpi, respectively). These observations suggest that while both Seg3 and Seg4 mRNAs are present in viroplasms, individual mRNA ratios vary between viroplasms even within the same cell.

## Rotavirus transcript association requires RNA chaperone NSP2

We hypothesised that the observed RNA assembly depends on the production of RNA-binding proteins that concentrate in viroplasms such as NSP2 *Jayaram et al., 2002*; *Viskovska et al., 2014* known to promote inter-molecular RNA association in vitro *Borodavka et al., 2017*; *Bravo et al., 2018*; *Bravo et al., 2021*. To investigate the role of NSP2 in the observed RNA assembly, we analysed two cell lines, each of which was expressing a short-hairpin RNA (shRNA) targeting either the NSP2 gene, or a scrambled control RNA. At 4 hpi, shRNA-mediated NSP2 knockdown resulted in an overall reduction of signal intensities for both Seg3 and Seg4 RNAs (*Figure 2—figure supplement 2*). Importantly, NSP2 knockdown (*Figure 2—figure supplement 2*) disrupted the apparent aggregation of Seg3 and Seg4 transcripts that no longer formed high-intensity RNA foci (*Figure 2—figure supplement 2*). Signal intensity analysis of Seg3 and Seg4 RNAs suggests that Seg3 transcripts appear to be more stable upon NSP2 depletion compared to Seg4 RNAs in RV-infected cells (*Figure 2—figure supplement 2*). Together, these data indicate that NSP2 expression is required for RV RNA clustering, further suggesting the role of NSP2 in the formation of higher order RNA assemblies. Although we observed a lack of transcript clustering in rotavirus-infected cells expressing shRNA targeting NSP2, it is important to note that the formation of viroplasms was also impaired under these conditions, supporting the essential role of NSP2 in viroplasm assembly. To directly visualise transcript oligomerisation, we also carried out super-resolution imaging of individual transcripts using DNA-based Point Accumulation for Imaging in Nanoscale Topography (DNA-PAINT) approach *Jungmann et al., 2014*; *Schnitzbauer et al., 2017*. Quantitative qPAINT analysis (*Jungmann et al., 2016*; *Figure 3*) of

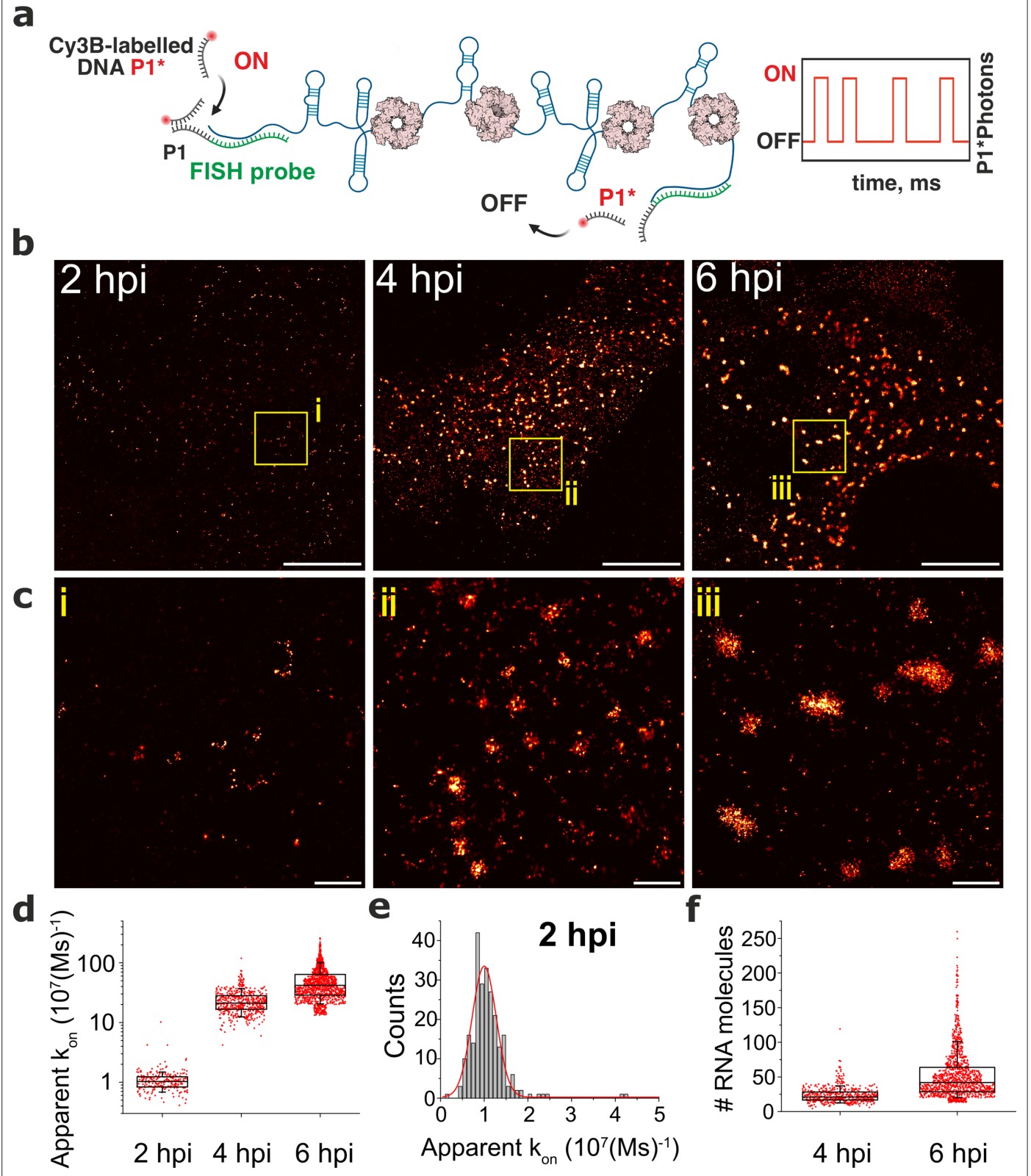

**Figure 3.** DNA-PAINT analysis of the RV RNA oligomerisation. (**a**) Schematics of DNA-PAINT imaging of RV transcripts. RNA hybridisation is used to install unique DNA 'docking' P1 strands for Seg3 RNA targets. These P1 sequences bind to complementary Cy3B-labelled DNA 'imager' strands (denoted P1*). Transient binding of P1* to accessible P1 sites on Seg3 RNAs is used for stochastic super-resolution reconstruction of the RNA targets. Only specific binding events (i.e. fully complementary DNA imager-docking strand) give rise to characteristic 'on/off' times, allowing discrimination from

*Figure 3 continued*

non-specific binding events. (**b**) DNA-PAINT super-resolution reconstruction of Seg3 transcripts at 2, 4, and 6 hpi. (**c**) Zoomed-in regions of Seg3 RNA structures highlighted in (**b**). (**d**) Quantitative PAINT (qPAINT) analysis of the RNA structures shown above. $k_{on}$ values were calculated for each selected structure assuming $k_{on} = (\tau_D \times c)^{-1}$, where $c$ is the imager strand concentration and $\tau_D$ the dark time between binding events ($c_{imager}$ = 1 nM at 2 hpi; 125 pM at 4 hpi and 100 pM at 6 hpi). Each point represents the apparent $k_{on}$ value for an RNA structure that is directly proportional to the relative number of the FISH probe binding sites, and thus reflects the number of transcripts within. (**e**) Unimodal distribution of $k_{on}$ values calculated for single Seg3 transcripts detected at 2 hpi. The resulting average $k_{on}$ value calculated from a Gaussian fit is $10^{-5}$ (Ms)$^{-1}$, corresponding to a single Seg3 transcript. (**f**) Estimated numbers of Seg3 transcripts (mean ± SD) at 4 hpi and 6 hpi are 24±12 ($\bar{x}$=21) and 53±36 ($\bar{x}$=42), respectively. N=245 (2 hpi), N=515 (4 hpi), N=1181 (6 hpi). Scale bars, 5 μm (**b**), 500 nm (**c**).

The online version of this article includes the following source data for figure 3:

**Source data 1.** The apparent $k_{on}$ value for RNA structures detected at 2, 4, and 6 hpi.

transcripts was used to assess the approximate number of its RNA-binding sites at 2 hpi (early infection stage) when the density of Seg3 transcripts is low. The qPAINT analysis (**Methods** and *Figure 3d*), revealed an apparent $k_{on}$ of $10^7$ (Ms)$^{-1}$ that corresponds to approximately ten smFISH probes per transcript *Jungmann et al., 2016*, consistent with these structures being single transcripts. Between 4 and 6 hpi, a fraction of Seg3 transcripts underwent assembly, yielding larger RNA clusters (*Figure 3*) that contained approximately 20–50 transcripts. Assuming that in viroplasms RNA target accessibility is expected to be lower than that of a single transcript, we note that the number of Seg3 transcripts in these structures is likely to be higher and using this approach transcripts are likely to be undercounted in larger granules. Nevertheless, these results indicate that transcript clustering is concomitant with the observed viral RNA aggregation during infection, and it occurs prior to the formation of detectable viroplasms.

## Single-cell Rotavirus transcriptome analysis using UDEx-FISH

To visualise the entire RV transcriptome in single cells, we developed and employed Universal DNA Exchange approach *Schueder et al., 2017* to combine it with smFISH (hereafter termed 'UDEx-FISH'). Eleven transcripts of interest were first pre-hybridised with each set of smFISH probes containing transcript-specific sequences that stably hybridise with the RNA, followed by a shorter a 'handle' sequence that binds fluorescently labeled complementary DNA probes ('Imager', *Figure 4a*). This approach allows installation of DNA 'handles' onto individual targets, thus enabling multiplexed imaging of targets irrespective of their molecular density unlike alternative combinatorial labelling schemes, for example, MERFISH *Chen et al., 2015*, and using spectrally similar dyes. Moreover, this approach does not require high concentrations of denaturants to remove pre-hybridized smFISH probes during sequential imaging previously used in MuSeq-FISH approach *Haralampiev et al., 2020*. Importantly, sequential imaging protocol minimizes RNA signal loss as bleached fluorophore-labelled probes are removed and replenished with a new imager after each round of visualisation, thus enabling accurate quantification of each transcript. To ascertain that during wash steps only DNA imagers were removed without the loss of transcript-specific smFISH probes, we carried out five iterative washes/imager applications. The relative fluorescence intensities of transcript-specific imagers remained unchanged (*Figure 4—figure supplement 1*) after five cycles of washes. More importantly, no residual signal was recorded after each individual wash step (*Figure 4—figure supplement 1*), and no transcript signal loss was observed due to bleaching. Finally, multiple rounds of washes did not alter the distribution of high-intensity RNA foci, nor had any apparent impact on the distribution or the morphology of RNA clusters (*Figure 4—figure supplement 1*), or EGFP-tagged viroplasms, confirming that the chosen approach and the designed probes were highly suitable for multiplexed characterisation of the RV transcriptome in single cells.

## Viroplasmic and cytoplasmic viral transcript stoichiometries are different

Using UDEx-FISH multiple copies of each transcript Seg1-Seg11 were readily detected (*Figure 4b*) at 2 hpi prior to the formation of viroplasms. Assuming similar rates of transcription for each individual genomic segment *Lu et al., 2008*; *Ayala-Breton et al., 2009*, transcription of longer segments is expected to yield fewer copies of longer Seg1-Seg4 transcripts (3.4–2.6 kb). However, Seg3 transcripts were more abundant compared to similarly sized Seg2 or Seg4 transcripts, suggesting that

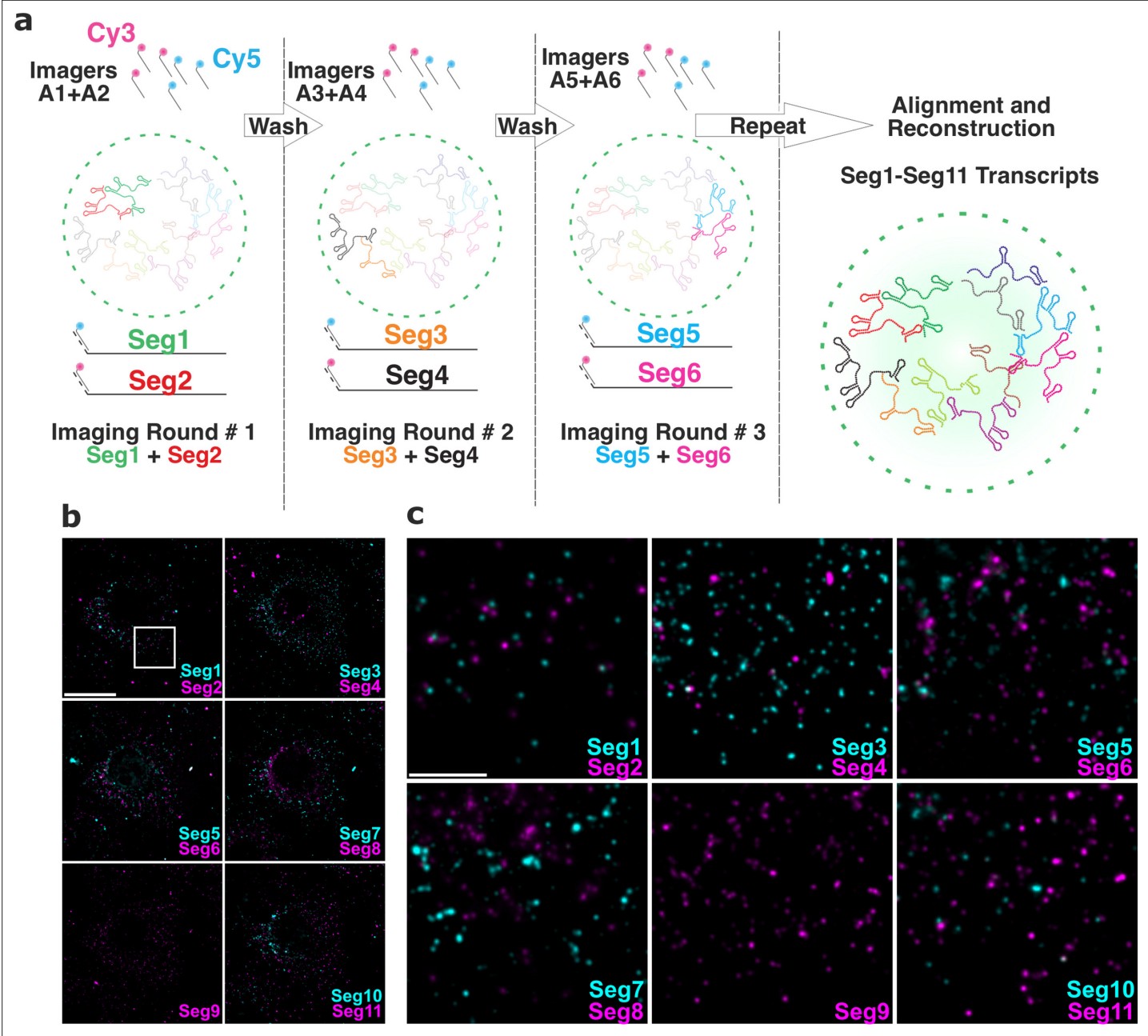

**Figure 4.** Universal DNA exchange-smFISH approach for single-cell imaging of the RV transcriptome. (**a**) Schematics of the Universal DNA Exchange-smFISH approach (UDEx-FISH). FISH probes targeting different RNA segments are extended with 12 nucleotide-long orthogonal DNA sequences $A_1$-$A_n$ (DNA 'handles'). Cy3 and Cy5-dye-modified DNA sequences (A1–An, 'Imagers') complementary to their respective handles can rapidly (<1 min) and stably hybridise with their segment-specific FISH probes RNA. Non-hybridised imagers are washed away and the two RNA targets, as well as NSP5-EGFP tagged viroplasms and nuclei are imaged in each imaging round (see Methods), allowing the next imaging round to take place. The imaging procedure is repeated until all targets of interest are recorded without affecting the quality of the fixed sample or loss of the RNA signal intensity due to photobleaching. (**b**) RV Transcriptome during early infection stage (2 hpi). Images were taken with same acquisition parameters for each imaging round. Scale bar, 20 μm. (**c**) Zoomed-in images of the highlighted areas shown in (**b**). Scale bar, 5 μm.

The online version of this article includes the following source data and figure supplement(s) for figure 4:

**Source data 1.** The integrated signal intensity after each successive round of imager dissociation and re-association.

**Figure supplement 1.** Validation of the UDEx-FISH approach in RV-infected MA104 cells.

individual segments may have different transcription rates or have different transcript stabilities in cells.

Next, we examined the RV transcriptome during the formation of viroplasms. At 4 hpi, EGFP-tagged viroplasms contained all 11 types of RV transcripts (*Figure 5*). At 6 hpi, UDEx-FISH quantification of RV transcripts in RV-infected cells was broadly in agreement with the RNA-Seq results (*Figure 5—figure supplement 1*) that also revealed that the RV transcriptome represented approximately 17% of all protein-coding transcripts in cells (*Figure 5b* and *Supplementary file 2*). Overall, the longest Seg1 (3.4 kb) and Seg2 (2.7 kb) transcripts were the least abundant species quantified by RNA-Seq and UDEx-FISH, followed by Seg5 (1.6 kb) suggesting that there is no simple correlation between the size of each transcript and its accumulation in cells (*Figure 5b and c*). Moreover, transcript stoichiometries in the cytoplasm and viroplasms were different (*Figure 5c and d*), and despite its lowest abundance overall, the largest Seg1 RNAs were the most enriched RNA species in viroplasms (*Figure 5d*). Similarly, the shortest Seg10 and Seg11 transcripts (0.67–0.75 kb) efficiently partitioned to viroplasms (*Figure 5d*), suggesting that RNA partitioning to these granules does not simply reflect GC content *Standart and Weil, 2018*, abundance or size[21,32,47] of transcripts.

To support genome assembly, viroplasms are expected to contain all 11 RV transcripts; we therefore also analysed transcript stoichiometries in individual viroplasms at 6 hpi (N=21) by comparing their relative intensities (abundance) to those of Seg1 transcripts (*Figure 5—figure supplement 1b*). As expected, all 11 transcripts were detected in all viroplasms. However, viroplasmic transcript stoichiometries deviated drastically for Seg7 ($R^2$=0.41), followed by Seg10 ($R^2$=0.67), Seg8 and Seg 11 ($R^2$=0.74), with the remaining transcripts having correlation coefficients in the range of $R^2$=0.79–0.92.

## 3' UTRs are required for RNA partitioning to viroplasms

Rotavirus transcript 3' untranslated regions (UTRs) are required for binding of the viral RNA polymerase VP1 *Patton and Chen, 1999*; *Patton, 1996* that has nanomolar affinity for the viroplasmic scaffold protein NSP5 *Arnoldi et al., 2007* and localizes to viroplasms *McKell et al., 2017*. We hypothesised that RNA partitioning to viroplasms may be governed by VP1 binding to the conserved UTR sequences. We chose Seg1 RNA due to its highest efficiency to partition in viroplasms (*Figure 5d*) and the shortest (17 nt) 3'UTR amongst all other RV transcripts. As described in Methods, we produced an EGFP-coding mRNA flanked by both UTRs derived from Seg1 transcript, as well as one lacking the 3'UTR denoted EGFP-Δ3'UTR (*Figure 6a*). The produced transcript was capped and electroporated into a previously described MA104 cell line constitutively expressing NSP2-mCherry that localises to viroplasms and enables their imaging (MA-NSP2-mCherry, see Methods). We chose electroporation as a delivery method to minimise RNA aggregation due to non-specific interactions with cationic lipids. Cells were infected 5 hr after RNA electroporation as described in Methods and fixed 5 hpi for FISH analysis of EGFP transcript localisation. Electroporation of the RNA construct lacking the 3'UTR (EGFP-Δ3'UTR, *Figure 6b*) did not yield any viroplasms containing Seg1 RNA signal in RV-infected cells (*Figure 6b*, top panel). A similar procedure using the transcript containing both UTRs resulted in RNA co-localisation with viroplasms (*Figure 6b*, bottom panel).

We also tested whether another viral transcript, NSP5-EGFP, containing the coding region of Seg11 RNA but lacking the 3'UTR would undergo enrichment in viroplasms during RV infection (*Figure 7a*). Again, NSP5-EGFP transcripts were diffusely distributed in the cytoplasm of RV-infected cells at 8 hpi. The presence of NSP5-EGFP-tagged viroplasms confirmed that NSP5-EGFP transcripts were functional (*Figure 7a*). However, devoid of segment-specific UTRs, these polyadenylated transcripts did not undergo enrichment in viroplasms (*Figure 7a*). To investigate whether the inclusion of a non-viral EGFP sequence might affect the localisation of a cognate viral RNA, we also visualised an EGFP-coding sequence fused to an NSP3-coding gene segment 7 that contained intact segment-specific UTRs (*Figure 7b*). These viral EGFP-coding transcripts accumulated in the NSP5-EGFP-tagged viroplasms produced in NSP3-2A-EGFP virus-infected cells (*Figure 7b*, the Pearson correlation coefficient, PCC, $R^2$=0.78 ± 0.1). Collectively, these results strongly suggest that segment-specific UTRs are important for the RV transcript localisation and enrichment in viroplasms.

Finally, since 3'UTRs are required for VP1 binding, we tested whether RNA accumulation in viroplasms is impaired when VP1 localisation to viroplasms is affected. We chose a well-characterised temperature-sensitive mutant C (tsC) with a single L138P substitution in VP1 that abrogates its ability to accumulate in viroplasms *Ramig, 1982* in a temperature-dependent manner *McKell et al., 2017*;

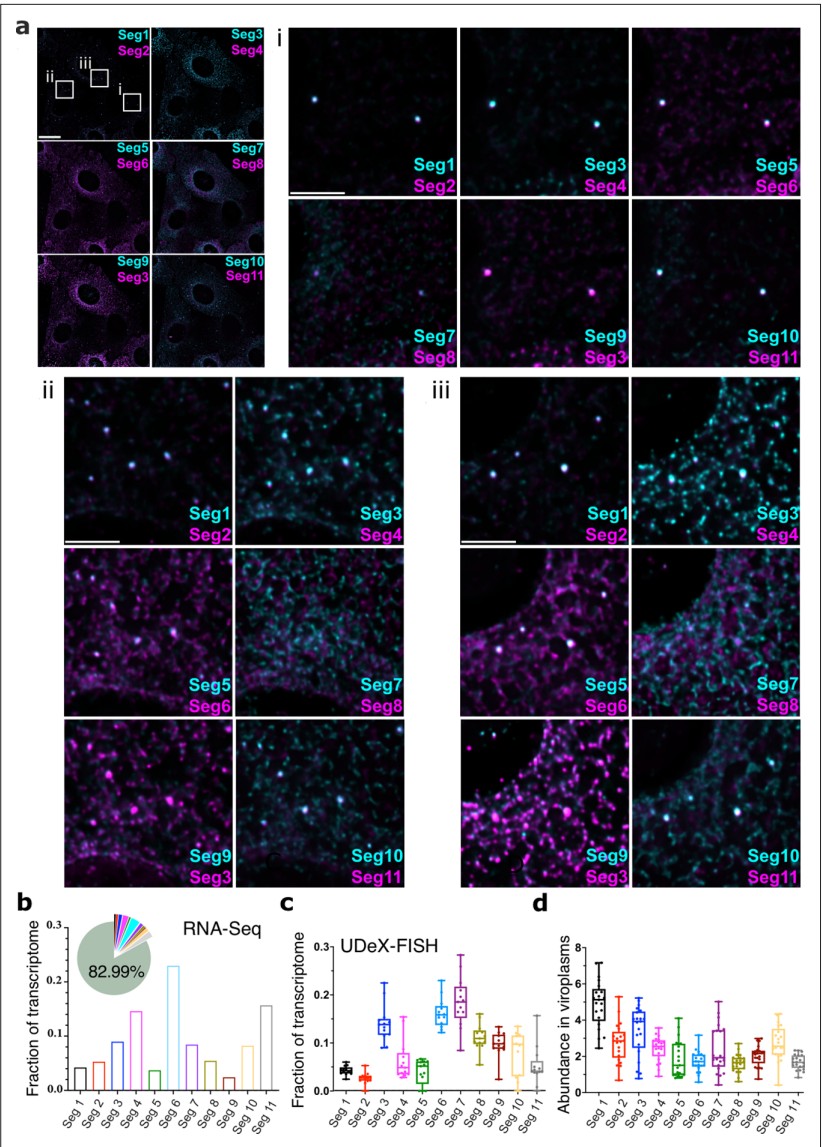

**Figure 5.** Single-cell analysis of the RV transcriptome. (**a**) In situ analysis of the RV transcriptome at 6 hpi. Scale bar, 20 μm. (**b**) Zoomed-in images of the highlighted regions containing viroplasms and 11 distinct RNA targets. Scale bar: 5 μm. (**b**) The proportion of RNA-seq reads assigned to the annotated protein-coding transcriptome of MA104 cells infected with RVA strain RF at 6 hpi. The RVA transcriptome represented ~17% of total reads (Methods), with the remaining 82.99% reads mapped onto MA104 protein-coding transcriptome. Bar plot shows the relative fractions of individual RV transcripts in the RV transcriptome at 6 hpi (Seg1-Seg11). (**c**) Single-cell analysis of the relative fractions of each segment-specific transcript (N=15 cells) estimated from the UDEx-FISH images data shown in panel (**a**). (**d**) Relative RV transcript abundance in viroplasms. Each data point represents the ratio of the viroplasmic signal density (per pixel) to the signal density of the corresponding cell. Higher values indicate stronger viroplasmic accumulation of viral transcripts. Lines, boxes, and whiskers represent the median, quartiles, and 5–95 percentile range of the data points, respectively. Relative fractions (mean ± SD) of individual viral transcripts detected in individual viroplasms (N=21).

The online version of this article includes the following source data and figure supplement(s) for figure 5:

**Source data 1.** Relative fractions of individual RV transcripts in the RV transcriptome at 6 hpi (Seg1-Seg11).

**Source data 2.** Single-cell analysis of the relative fractions of each segment-specific transcript (N=15 cells) estimated from the UDEx-FISH images data.

**Source data 3.** Relative abundance of individual transcripts in viroplasms (N=21) estimated by UDEx-FISH.

**Source data 4.** Quantitative comparison of the RV transcriptome estimated by UDEx-FISH and RNA-Seq at 6 hpi.

*Figure 5 continued on next page*

*Figure 5 continued*

**Source data 5.** Integrated signal intensities for individual viroplasm-localized RV transcripts used for correlation analysis of the RV RNA stoichiometry in viroplasms.

**Figure supplement 1.** Quantification of transcripts by UDEx-FISH.

*Nilsson et al., 2021*. At both permissive (31 °C) and non-permissive (39 °C) temperatures, viral transcripts accumulated in the cytoplasm of infected cells (*Figure 7c*), confirming that the L138P mutant retains its transcriptional activity at 39 °C. Similarly, viroplasms were formed at both temperatures. Yet, only at the permissive temperature viroplasms contained RV transcripts ($R^2$=0.62 ± 0.17), while at the non-permissive temperature RV transcripts clustered outside viroplasms ($R^2$=0.18 ± 0.07), indicating that RNA partitioning to viroplasms is linked to the ability of VP1 to localize to these granules. Together, these results suggest that RV RNA partitioning to viroplasms requires both 3' UTRs and viral polymerase VP1 localisation to these replication factories.

## Discussion

Ribonucleoprotein (RNP) granules formed via LLPS are ubiquitous in cells *Khong et al., 2017*; *Garcia-Jove Navarro et al., 2019*; *Langdon and Gladfelter, 2018*; *Roden and Gladfelter, 2021*; *Rhine et al., 2020*; however, each specific type of an RNP condensate is likely to be unique in protein and RNA composition that can be dynamically modulated in response to stimuli. While some key protein constituents of various RNP granules have been identified, their transcriptomes remain less well-characterised due to the granule isolation and purification challenges. Recently, we have shown that rotavirus viroplasms represent RNP granules initially formed via liquid-liquid phase separation (LLPS) of the RNA chaperone NSP2 and a condensate-forming protein NSP5 *Geiger et al., 2021*. Here, we provide the first glimpse at the unique RNA composition of the viral condensate that support replication of an 11-segmented RNA genome.

RNA-Seq and single-cell UDEx-FISH analyses have revealed that at 6 hpi, rotavirus infection resulted in non-stoichiometric accumulation of distinct viral transcripts, in agreement with previous bulk kinetic studies of RV replication carried out during late infection *Ayala-Breton et al., 2009*; *Patton and Spencer, 2000*; *Silvestri et al., 2004*. Early infection (2 hpi) RV transcriptome analysis by UDEx-FISH shows that all 11 types of the RV transcripts were diffusely distributed in the cytoplasm of infected cells. At 6 hpi, Seg1 (VP1-coding) transcripts represented the smallest fraction of the RV transcriptome. While RNA-Seq quantification provided the relative ratios of viral and non-viral protein-coding transcripts, it failed to reveal specific RNA localisation and subcellular distribution in individual RV-infected cells. In contrast, our UDEx-FISH analysis appeared to undercount smaller RNA targets (Seg11 and Seg10), presumably due to a limited number of smFISH probes successfully hybridized to these transcripts. We therefore used both approaches to quantify the viral transcriptome in RV-infected cells. The initial transcription stage was followed by the apparent formation of higher order assemblies of the RV transcripts during a process that required the viral RNA chaperone NSP2. Our observations are fully consistent with previously reported inhibition of genome replication and virion assembly resulting from the loss of NSP2 expression *Silvestri et al., 2004*, providing new evidence for NSP2 as a multivalent RNA chaperone that links together populations of viral transcripts. While NSP2 has been demonstrated to promote RNA oligomerisation in vitro, the shRNA-mediated knockout of NSP2 may also disrupt RNA distribution and localisation due to the impaired viroplasm formation, suggesting a potential role for NSP2 in these processes. In light of our recent findings *Geiger et al., 2021*, these results reveal several parallels between viroplasmic condensates and other cytoplasmic RNP granules, including stress granules (SGs) and P-bodies. Both SGs and viroplasms represent liquid-like RNP assemblies that form from untranslating mRNAs *Khong et al., 2017*; *Van Treeck and Parker, 2018*; *Wheeler et al., 2016*. Expression of SG-specific or viroplasm-specific multivalent RNA-binding proteins is essential to their formation. While many aspects of viroplasm formation mimic those seen in assembly of stress granules *Khong et al., 2017*; *Van Treeck and Parker, 2018*; *Wheeler et al., 2016*; *Tauber et al., 2020*, RNA partitioning into viroplasms and the observed clustering of transcripts appears to be virus-specific, implying that both processes depend on cognate, RV-specific RNA-protein and RNA-RNA interactions. Unlike SGs, whose RNA composition is biased toward larger, AU-rich mRNAs *Khong et al., 2017*, our data reveal that viroplasmic RNA enrichment is likely to be

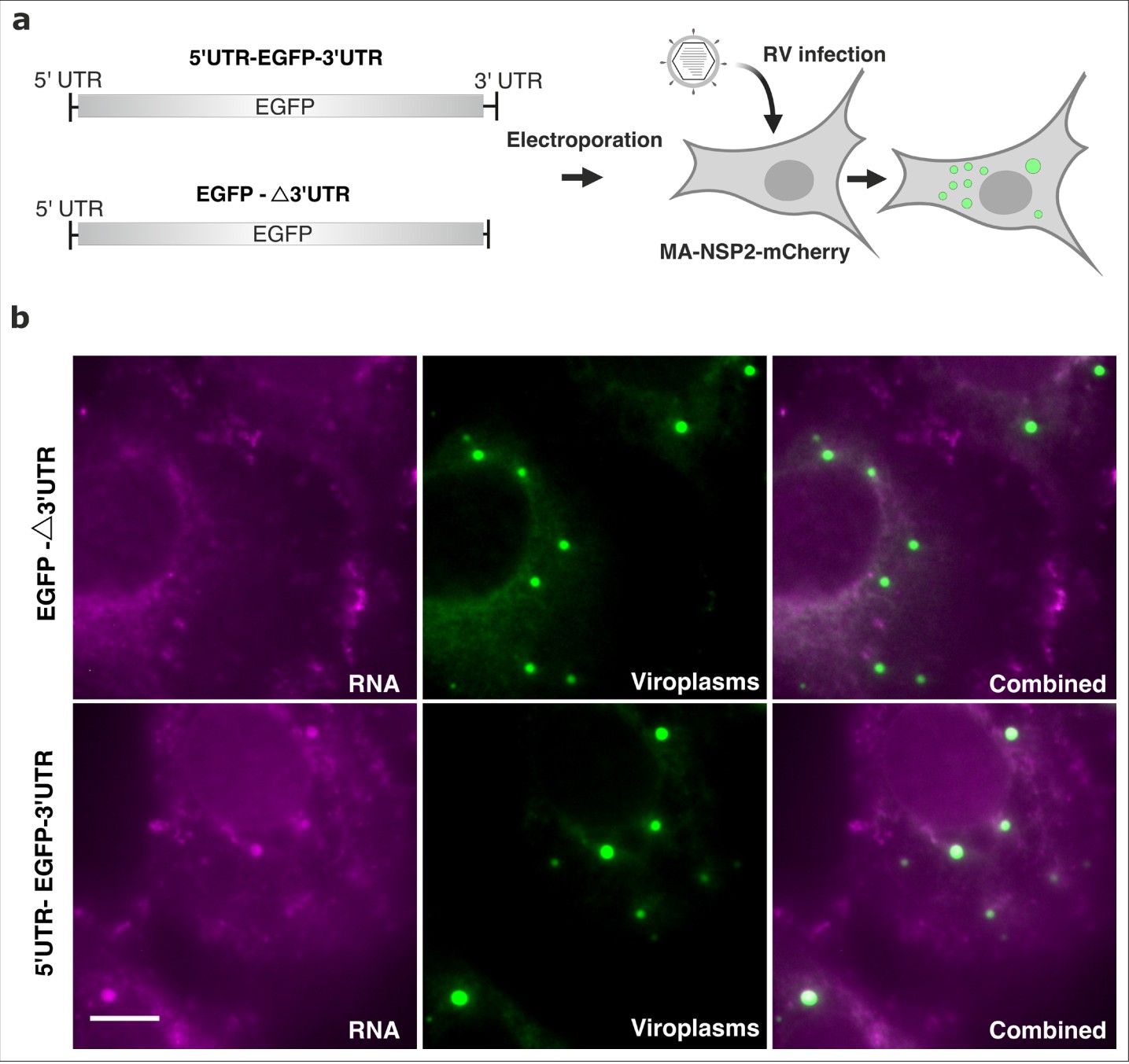

**Figure 6.** 3' UTR is required for RV transcript localisation to viroplasms. (**a**) Schematics of the EGFP transcript construct containing the EGFP-coding sequence flanked by the 5'UTR and 3'UTR sequences derived from Seg1 RNA (5'UTR-EGFP-3'UTR) or only 3'UTR sequence (denoted EGFP-Δ3'UTR) that lacks the 17-nt long 3'UTR. Both RNA constructs were electroporated into MA104-mCherry-NSP2 cells 6 hr prior to being infected with RVs, as described in Methods. (**b**) Widefield images of RV-infected MA104-mCherry-NSP2 cells electroporated with EGFP-Δ3'UTR (top) and the 5'UTR-EGFP-3'UTR transcript (bottom). Cells were fixed at 6 hpi prior to the hybridisation with FISH probes against EGFP coding region, as described in Methods. Green: NSP2-mCherry-tagged viroplasms, magenta: smFISH for EGFP RNA. Scale bars, 10 μm.

determined by transcript-specific terminal sequences of each RV gene segment. It remains to be determined how the individual sequences within the transcript-specific UTRs of different length and composition determine RNA partitioning to these condensates.

A plausible model for such enrichment is based on a high affinity, specific protein-RNA recognition, such as previously reported conserved interaction between the 3' terminal sequence of each RV transcript and its RNA-dependent RNA polymerase (RdRP) VP1 that exhibits high affinity for NSP5

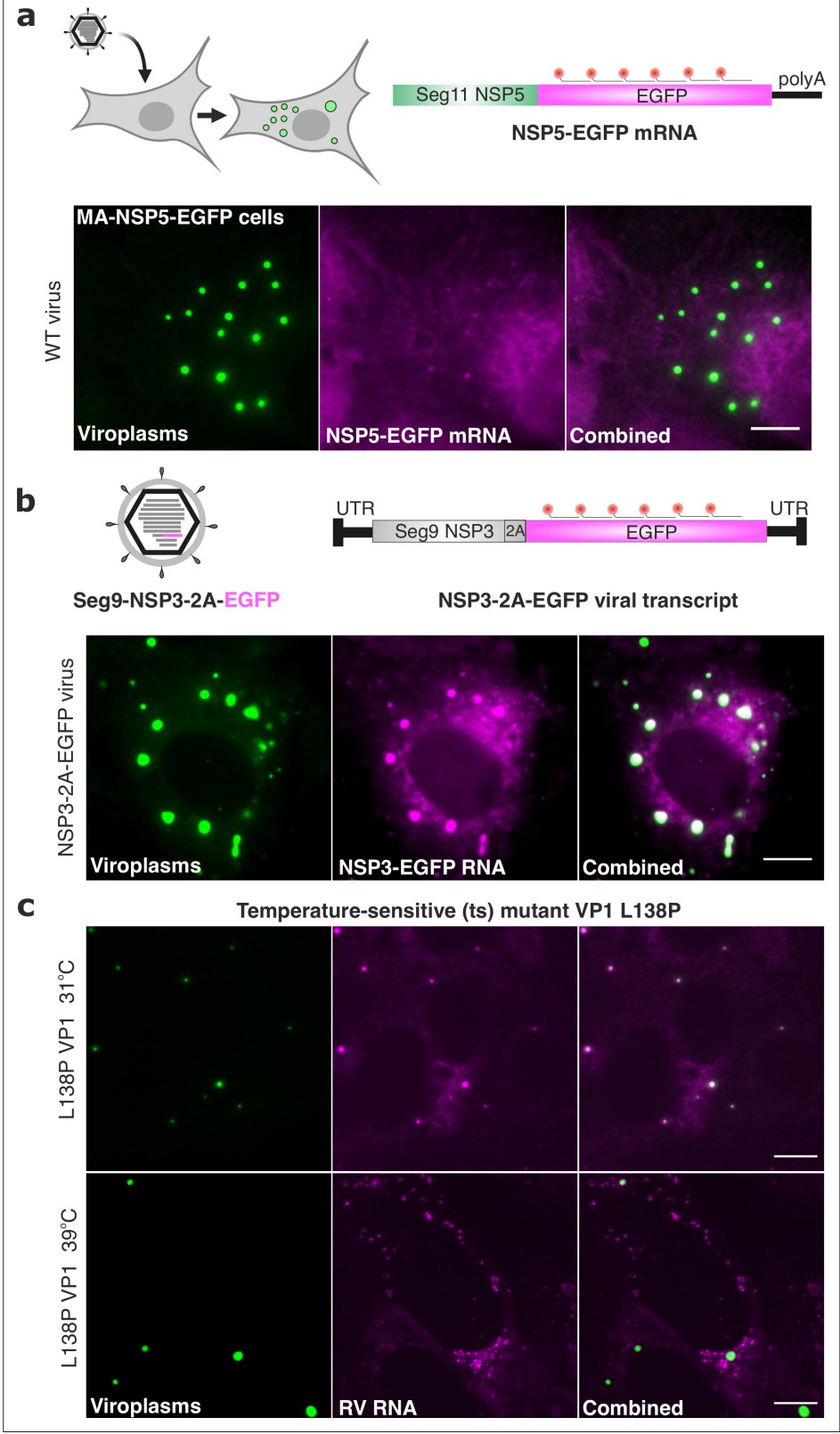

**Figure 7.** 3′ UTRs and the viral polymerase are required for RV transcript localisation. (**a**) Polyadenylated RV mRNAs coding EGFP do not partition to viroplasms. Top - experimental design of the smFISH probes to target RV mRNAs coding EGFP in cells infected with the wild-type rotavirus. Bottom – smFISH of NSP5-EGFP-coding transcripts in RV-infected cells at 8 hpi. (**b**) Top - experimental design of smFISH probes to target the EGFP-

*Figure 7 continued on next page*

*Figure 7 continued*

coding RNAs produced by the recombinant RV (rRV, denoted as NSP3-2A-EGFP containing Seg7-specific UTRs). Bottom - smFISH of NSP3-2A-EGFP transcripts in rRV-infected cells at 8 hpi. Note the colocalisation of NSP3-2A-EGFP transcripts (magenta) with EGFP-tagged viroplasms (green). (**c**) L138P mutation affects the ability of VP1 to accumulate in viroplasms at 39 °C but not at 31 °C. smFISH of RV transcripts in cells infected with L138P mutant at 8 hpi grown at 39 °C vs 31 °C. Note the formation of EGFP-tagged viroplasms at both temperatures albeit RV transcripts efficiently accumulate in viroplasms at 31 °C (PCC, $R^2$=0.62 ± 0.17 vs 0.18±0.07 for 31 °C vs 39 °C, respectively). Errors represent standard deviations for N=4 analyzed ROIs. Images represent maximum intensity Z-projections. Scale bars, 10 μm.

*Patton and Chen, 1999*; *Patton, 1996*; *Arnoldi et al., 2007*. We propose that RdRP-binding sites within the 3' UTRs of RV transcripts would facilitate their enrichment in viroplasms. Interestingly, a similar RdRP-facilitated viral transcript selection mechanism was recently alluded to in SARS-CoV-2 RNA-nucleoprotein-rich condensates *Savastano et al., 2020*, suggesting that selective viral transcript enrichment in replicative condensates may be widely employed by other RNA viruses. Our model (*Figure 8*) also accounts for the accumulation of VP1-bound nontranslating viral transcripts in viroplasms, in which the formation of inter-molecular RNA-RNA interactions between Seg1-Seg 11 transcripts is favoured in the presence of the viral RNA chaperone NSP2 *Borodavka et al., 2018*; *Borodavka et al., 2017*; *Bravo et al., 2021*.

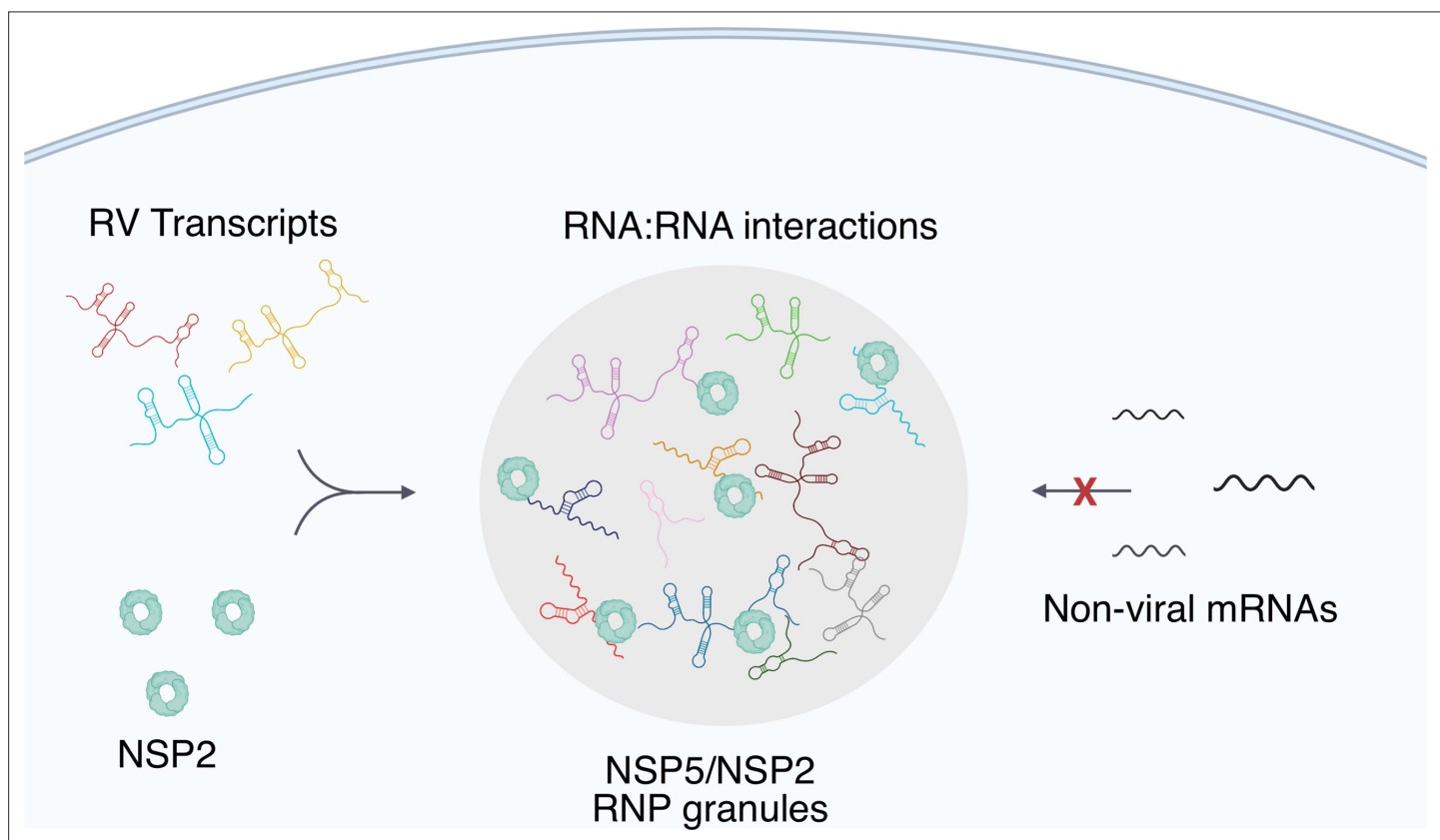

**Figure 8.** A proposed model of viral transcript partitioning into viroplasms. RV transcripts (for clarity, only 3 out of 11 distinct transcripts are shown) associate with viral RNA-binding proteins NSP2 (cyan doughnut-like octamers), as well as the RV RNA-dependent RNA Polymerase VP1 *Viskovska et al., 2014*; *Patton, 1996* (not shown). These transcripts can partition into NSP5/NSP5-rich droplets, forming viroplasms containing all 11 distinct types of RV transcripts. While the number of individual types of transcripts varies, their relative numbers in viroplasms is close to equimolar, suggesting an RNA partitioning mechanism distinct from the formation of stress granules or P-bodies. High effective concentration of cognate RNAs within viroplasms is conducive to the formation of inter-molecular RNA-RNA interactions between 11 distinct RV transcripts required for their stoichiometric assembly. Given the degenerative nature of RNA-RNA base-pairing *Van Treeck and Parker, 2018*, viroplasms must maintain their RNA composition to promote inter-molecular interactions between the RV transcripts, whilst minimising the partitioning of other non-cognate mRNAs that may interfere with segmented genome assembly.

RNA-binding proteins NSP2 and NSP5 are the most abundant viral proteins produced during early stages of RV infection *Patton et al., 2006*, each being indispensable for viral RNP condensate formation. The $K_d$ value of the promiscuous RNA-binding protein NSP2 for ssRNA is low nanomolar *Bravo et al., 2018*; *Hu et al., 2012*, therefore protein-free RNAs would be expected to be bound by it, consistent with our previous observations of RV transcripts interacting with a non-viral RNA in the presence of NSP2 in vitro *Borodavka et al., 2017*. Given the degenerative nature of RNA-RNA base-pairing, such non-specific interactions would be expected to interfere with the stoichiometric assembly of viral transcripts. Thus, such transcript sequestration within the specialized RNP granules offers a solution to the problem of RNA assortment which is likely governed by inter-molecular RNA interactions and must avoid spurious non-cognate RNA-RNA interactions. We propose that viroplasms may act as the crucibles for the assortment of RV transcripts, and further studies to explore the exact mechanisms of RNA enrichment in these granules will underpin search for new antiviral strategies.

# Methods

## Key resources table

| Reagent type (species) or resource | Designation | Source or reference | Identifiers | Additional information |
|---|---|---|---|---|
| Cell line (*Chlorocebus aethiops*) | MA104 Clone 1 | ATCC | ATCC CRL-2378.1; RRID:CVCL_3845 | Stable cell line |
| Cell line (*Chlorocebus aethiops*) | MA104-NSP5-EGFP | Cells obtained from Dr. O. Burrone and G. Papa, ICGEB, Trieste, Italy | https://doi.org/10.1128/JVI.01110-19 | Stable cell line |
| Cell line (*Chlorocebus aethiops*) | MA104-shRNA-NSP2 | This manuscript | MA104-shRNA-NSP2 | Made by transfection with pPB[shRNA]-EGFP:T2A:Puro-U6 plasmid |
| Cell line (*Chlorocebus aethiops*) | MA104-NSP2-mCherry | *Geiger et al., 2021* | https://doi.org/10.15252/embj.2021107711 | Stable cell line |
| Strain, strain background (Group A rotavirus) | Bovine rotavirus group A, strain RF [G6P6(1)] | Dr Ulrich Desselberger, University of Cambridge, UK | https://doi.org/10.1371/journal.pone.007432 | |
| Strain, strain background (Group A rotavirus) | Simian rotavirus group A, strain SA11/tsC mutant | Dr Sarah McDonald Esstman, Wake Forest University, USA | https://doi.org/10.1016/j.virusres.2021.198488 | |
| Sequence-based reagents | FISH probes | Integrated DNA Technologies | RNA FISH DNA probes | Sequences listed in *Supplementary file 1* |
| Antibody | Goat polyclonal, HRP-conjugated, anti-guinea pig | ThermoFisher | A18775, RRID:AB_2535552 | 1:10,000 dilution used for Western Blotting |
| Antibody | Anti-NSP2 (polyclonal, anti-guinea pig) | Dr. O. Burrone, ICGEB, Trieste, Italy; *Papa et al., 2019* | https://doi.org/10.1128/JVI.01110-19 | 1:1,000 dilution used for Western Blotting |
| Sequence-based reagents | FISH Probes, fluorescently labelled Seg3RF-Q670 Seg4RF-Q570 GAPDH-Q670 Comb-Q570 For_UTR Rev_UTR Rev_dUTR | LGC Biosearch Technologies | Seg3RF-Q670 Seg4RF-Q570 GAPDH-Q670 Comb-Q570 | Sequences provided in the *Supplementary file 1* |
| Other | Stellaris RNA FISH Hybridization Buffer | LGC Biosearch Technologies | SMF-HB1-10 | |
| Recombinant DNA reagent | pPB[shRNA]-EGFP:T2A:Puro-U6 | VectorBuilder https://en.vectorbuilder.com | pPB[shRNA]-EGFP:T2A:Puro-U6 | Sequence provided in *Supplementary file 1* |

*Continued on next page*

*Continued*

| Reagent type (species) or resource | Designation | Source or reference | Identifiers | Additional information |
|---|---|---|---|---|
| Recombinant DNA reagent | pCMV-HyPBase | Dr. O.Burrone, ICGEB, Trieste, Italy; **Papa et al., 2019** | https://doi.org/10.1128/JVI.01110-19 | |
| Recombinant DNA reagent | pT7-NSP3-EGFP | Dr. O.Burrone, ICGEB, Trieste, Italy; **Papa et al., 2019** | https://doi.org/10.1128/JVI.01110-19 | |
| Commercial assay or kit | NEBNEXT rRNA Depletion kit | New England Biolabs | E6350S | Mouse/Rat/Human species |
| Commercial assay or kit | Faustovirus Capping Enzyme | New England Biolabs | M2081S | |
| Commercial assay or kit | mRNA Cap 2′-O-methyltransferase | New England Biolabs | M0366 | |
| Other | Lipofectamine 3000 | Invitrogen | L3000001 | |
| Commercial assay or kit | MycoSPY | Biontex Laboratories, Germany | M030-050 | PCR detection of *Mycoplasma sp.* |
| Commercial assay or kit | NEBNEXT Ultra II FS DNA Library Prep kit | New England Biolabs | E7805S | |
| Commercial assay or kit | RNA extraction kit, RNEasy | Qiagen | 74034 | |
| Other | SuperScript II reverse transcriptase | Invitrogen | 18064014 | |
| Commercial assay or kit | HighScribe T7 kit | New England Biolabs | E2040S | |
| Other | DAPI stain | Invitrogen | D1306 | |
| Other | Atto647N-Oligo(dT30) | Integrated DNA Technologies | https://doi.org/10.1016/j.molcel.2017.10.015 | |
| Software, algorithm | OriginPro | OriginLab | RRID:SCR_014212 | |
| Software, algorithm | Icy | (http://icy.bioimageanalysis.org/) | RRID:SCR_010587 | Colocalisation analysis |
| Software, algorithm | Picasso | Developed in-house (**Kowalewski, 2023**) https://github.com/jungmannlab/picasso | Schnitzbauer et al., *Nat. Protoc.* **12**, 1198–1228 (2017). | |
| Software, algorithm | Stellaris Designer | LGC Biosearch Technologies | https://www.biosearchtech.com/stellaris-designer | RNA FISH probe designer |
| Software, algorithm | Salmon | https://github.com/COMBINE-lab/salmon (**COMBINE lab, 2022**) | Patro, R. et al., *Nat. Methods* **14**, 417–419 (2017). | |

## Cells and viruses

MA104 (Clone 1, ATCC CRL-2378.1) cells were cultivated as previously described *Arnold et al., 2012*. This clone was obtained directly from ATCC. MA104 cell line (*Cercopithecus aethiops kidney epithelial cells*) stably expressing NSP5-EGFP *Eichwald et al., 2004* was cultured in DMEM (Dulbecco's modified Eagle medium, GlutaMax-I, 4.5 g/L glucose, ThermoFisher), supplemented with 10% foetal bovine serum (FBS), 1% non-essential amino acids solution (Sigma), 1 mM sodium pyruvate (Sigma) and 500 μg/ml G418 (Roche). MA104 cell line (*Cercopithecus aethiops kidney epithelial cells*) stably expressing NSP2-mCherry *Eichwald et al., 2004* was cultured as NSP5-EGFP cell line. Rotavirus A (RVA) strain RF (G6P6[1]) was a generous gift from Dr Ulrich Desselberger (University of Cambridge, UK). It was cultivated, harvested and stored, as described previously *Arnold et al., 2012*; *Cheung et al., 2010*. For RNA imaging experiments, MA104 cells and their derivatives (MA104-NSP5-EGFP),

MA104-shRNA-NSP2 were seeded into Ibidi 8-well μ-slides and allowed to reach 90% confluency prior to the infection. Confluent cell monolayers were rinsed twice with DMEM medium without FBS for 15 min to remove any residual FBS, and were subsequently infected with trypsin-activated rotavirus stocks, as described in *Arnold et al., 2012* at multiplicity of infection (MOI) of 10. Cells were fixed at different time points of infection, as described below. All cell lines were PCR-tested for *Mycoplasma sp.* contamination (MycoSPY, Biontex Laboratories, Germany).

## Generation of stable cell lines

MA104-shRNA-NSP2 cell line was generated using the PiggyBac system. Briefly, $10^5$ MA104 cells were co-transfected with the plasmid pCMV-HyPBase encoding the hyperactive variant of PiggyBac transposase *Yusa et al., 2011*; *Li et al., 2013*; *Hubstenberger et al., 2017* along with the plasmid pPB[shRNA]-EGFP:T2A:Puro-U6 harbouring shRNA targeting RVA NSP2 gene using Lipofectamine 3000 (Sigma-Aldrich), following the manufacturer's instructions. The cells were maintained in DMEM supplemented with 10% FBS for 3 days, and then the cells were subjected to selection in the presence of 5 μg/ml puromycin (Sigma-Aldrich) for 4 days, prior to further selection by FACS sorting for EGFP expression.

## Western blot analysis

Proteins were treated with SDS and 2-mercaptoethanol at 95 °C, resolved by SDS-PAGE on Tris-glycine gels and transferred onto nitrocellulose membranes. Afterwards, the membranes were blocked with PBS containing 5% (w/v) skimmed milk and 0.1% Tween-20 and then incubated with guinea pig NSP2-specific antibody diluted 1:1,000 in PBS containing 1% milk and 0.1% Tween-20. Blots were washed with PBS and incubated with HRP-conjugated secondary antibodies in PBS (1:10,000) containing 1% milk and 0.1% Tween-20. Blots were developed with SuperSignal West Pico Chemiluminescent Substrate (Pierce) and exposed to BioMax MR film (Kodak). All scanned images were post-processed in Adobe Illustrator.

## Recombinant NSP3-2A-EGFP virus

Rescue of the recombinant NSP3-2A-EGFP virus (strain SA11) was carried out as previously described *Papa et al., 2020*; *Papa et al., 2019*; *Philip et al., 2019*; *Philip and Patton, 2020*. Briefly, monolayers of BHK-T7 cells ($4\times10^5$) cultured in 12-well plates were co-transfected using 2.5 μL of TransIT-LT1 transfection reagent (Mirus) per microgram of DNA plasmid. Each mixture comprised 0.8 μg of SA11 rescue plasmids: $pT_7$-VP1, $pT_7$-VP2, $pT_7$-VP3, $pT_7$-VP4, $pT_7$-VP6, $pT_7$-VP7, $pT_7$-NSP1, $pT_7$-NSP3-2A-EGFP, $pT_7$-NSP4, and 2.4 μg of $pT_7$-NSP2 and $pT_7$-NSP5. 0.8 μg of pcDNA3-NSP2 and 0.8 μg of pcDNA3-NSP5, encoding NSP2 and NSP5 proteins, were also co-transfected to increase the efficiency of virus rescue. At 24 hr post-transfection, MA104 cells ($5\times10^4$ cells) were added to transfected cells. The cells were co-cultured for 3 days in FBS-free medium supplemented with porcine trypsin (0.5 μg/mL) (Sigma Aldrich). After incubation, transfected cells were lysed by freeze-thawing and 0.2 ml of the lysate was used to infect fresh MA104 cells. After adsorption at 37 °C for 1 hr, cells were washed three times with PBS and further cultured at 37 °C for 4 days in FBS-free DMEM supplemented with 0.5 μg/mL trypsin (Sigma Aldrich, 9002-07-7) until a clear cytopathic effect was visible. Successful production of EGFP by the recombinant virus was confirmed microscopically.

## Single-molecule fluorescence in situ hybridisation (smFISH)

Rotavirus-infected and mock-infected MA104 cell controls, where appropriate, were fixed with 4% (v/v) methanol-free paraformaldehyde in nuclease-free phosphate saline buffer (PBS) for 10 min at room temperature. Samples were then washed twice with PBS, and fixed cells were permeabilised with 70% (v/v) ethanol (200 proof) in RNAse-free water, and stored in ethanol at +4 °C for at least 12 hr prior to hybridisation, and no longer than 24 hr. Permeabilized cells were then re-hydrated for 5 min in a pre-hybridisation buffer (300 mM NaCl, 30 mM trisodium citrate, pH 7.0 in nuclease-free water, 10 % v/v Hi-Di formamide (Thermo Scientific), supplemented with 2 mM vanadyl ribonucleoside complex). Re-hydrated samples were hybridized with an equimolar mixture of DNA probes specific to the mRNA targets (RVA RF or *C.aethiops* GAPDH transcripts), 62.5 nM final concentration, see *Supplementary file 1*, in a total volume of 200 μl of the hybridisation buffer (Stellaris RNA FISH hybridisation buffer, Biosearch Technologies, supplemented with 10% v/v Hi-Di formamide). After 4 hr of incubation at

37 °C in a humidified chamber, samples were briefly rinsed with the wash buffer 300 mM NaCl, 30 mM trisodium citrate, pH 7.0, 10 % v/v formamide in nuclease-free water, after which a fresh aliquot of 300 μl of the wash buffer was applied to each well and incubated twice at 37 °C for 30 min. After three washes, nuclei were briefly stained with 300 nM DAPI solution in 300 mM NaCl, 30 mM trisodium citrate, pH 7.0) and the samples were finally rinsed with and stored in the same buffer without DAPI prior to the addition of photostabilising imaging buffer (PBS containing an oxygen scavenging system of 2.5 mM protocatechuic acid, 10 nM protocatechuate-3,4-dioxygenase supplemented with 1 mM (±)–6-hydroxy-2,5,7,8-tetramethylchromane-2-carboxylic acid (Trolox) *Aitken et al., 2008*.

Sequences of the oligonucleotide RNA FISH probes, as listed in *Supplementary file 1* were used. These were generated using the Stellaris RNA FISH probe designer (https://www.biosearchtech.com/stellaris-designer), using each gene-specific sequences (see *Supplementary file 1* for GenBank IDs) and level 2 masking. The resulting pools of probes were then further filtered to remove the sequences targeting the RNA transcripts sequences with higher propensity to form stable intra-molecular base-pairing. Oligo(dT) FISH was carried out as above, except a 3'-ATTO647N-dye labelled HPLC-purified 30-mer oligo-dT (IDT) was used instead of pooled RV-specific FISH probes.

## Universal DNA exchange FISH (UDEx-FISH)

For UDEx-FISH, sequences of the oligonucleotide RNA FISH probes are listed in *Supplementary file 1*. Gene-specific portions of each probe were generated using the Stellaris RNA FISH probe designer, followed by a TT linker and a 10-nt long DNA handle designed to minimise any potential intra-molecular base-pairing *Schueder et al., 2017*. All FISH hybridisation steps were identical to the ones described in the section above, except the individual target-specific probe concentration was adjusted to 62.5 nM in the hybridisation mix. Cy3- and Cy5-modified labelling DNA strands ('Imagers') were incubated in the labelling buffer (600 mM NaCl, 2.7 mM KCl, 8 mM $Na_2HPO_4$ and 2 mM $KH_2PO_4$ pH 7.4 in nuclease-free water), for approximately 5 min. Samples were rinsed with the labelling buffer, followed by addition of the imaging buffer (*vide supra*). After a round of image acquisition, hybridized labelling strands were dissociated by briefly incubating samples in 30% (v/v) formamide in PBS ('Wash buffer') for 2–3 min at RT, repeating this procedure twice, whilst monitoring for any residual fluorescence signal in four acquisition channels (Cy5, Cy3, GFP, DAPI). Formamide-containing wash buffer was then aspirated, and samples were rinsed twice with fresh labelling buffer prior to the introduction of the next batch of 'Imager' DNA strands, thus concluding a single imaging cycle. To calibrate the recorded fluorescence intensities to quantify the signals originating from the individual transcripts in an unbiased manner due to differences in properties of spectrally distinct dyes (Cy3 vs Cy5), we swapped Cy5 and Cy3-dye-labelled DNA imagers for one of the RNA targets in each imaging cycle. A total number of six imaging cycles were required to image the RV transcriptome, and the last cycle always contained an Imager targeting the segment imaged during cycle 1 to control for a possibility of the signal loss due to the dissociation of FISH probes, and for signal calibration purposes.

## RNA production and RNA delivery

A pT7-NSP3-EGFP construct (see Key Resources Table) was used for amplification of EGFP ORF-containing transcription templates using a Q5 High-Fidelity DNA Polymerase with the forward primer (For_UTR) and either Rev_UTR (to make the 5'UTR-EGFP-3'UTR DNA template) or Rev_dUTR (to produce the EGFP-3'ΔUTR DNA template). DNA templates were purified using Monarch PCR purification kit New England Biolabs following the manufacturer's protocol. Purified DNA templates were used for run-off transcription with T7 polymerase-based HiScribe kit (NEB) following the manufacturer's recommendations to generate 5'UTR-EGFP-3'UTR and EGFP-3'ΔUTR transcripts, as described in *Coria et al., 2022* Transcripts were purified using RNEasy kit (Qiagen), and capped using one-pot capping reaction with Faustovirus capping enzyme and mRNA Cap 2'-O-Methyltransferase (New England Biolabs) following the manufacturer's protocol. Capped transcripts were purified with RNEasy kit and quantified spectrophotometrically prior to the electroporation. For electroporation, MA104-NSP2-mCherry cells were harvested at 80% confluency by trypsinisation, and collected by centrifugation (1000xg, 3 min). Cells were resuspended in 1 ml Gibco Opti-MEM medium (Fisher). 0.15 ml of the cell suspension was mixed with 100 0.1 ml of Opti-MEM containing 3 μg of the RNA. The mixture was transferred into electroporation cuvettes (Fisher) with a 2 mm gap (Fisher #FB102). Electroporation was carried out using a NEPA21 Electroporator Type II (Poring pulse: 175 V, length: 2.5ms, interval:

50ms, Decay: 10%; Transfer pulse: 30 V, length: 50ms, interval: 50ms, Decay: 40%). One ml of the complete DMEM supplemented with 10% FBS was added, and cells (0.3 ml/well) were seeded into Ibidi 8-well μ-slides 5–6 hr before infection with RVs. Cells were infected with the WT virus at MOI = 10. At 6 hpi, cells were fixed and prepared for FISH analysis and imaging, as described above.

## Image data acquisition

Widefield imaging was carried out on a Leica (Wetzlar, Germany) DMI6000B inverted microscope equipped with a LEICA HCX PL APO 63 x/NA1.4 oil immersion objective. Dye excitation was performed with a cooled pe-4000 LED source illumination system at the wavelengths 385 nm (DAPI), 470 nm (EGFP), 550 nm (Cy3 and Quasar 570), and 635 nm (Cy5 and Quasar 670). Fluorescent signals were detected with a Leica DFC9000 GT sCMOS camera with a pixel size of 6.5 μm. Images were acquired over a full field of view of the camera chip (2048×2048 pixels) resulting in a total imaging region of 211 μm × 211 μm. Exposure times were adjusted accordingly to the signal intensity to avoid pixel saturation. Typical exposure times were 250ms for DAPI, 500ms for EGFP, 750ms – 1 s for Cy3/Cy5 or Quasar 570/670 dyes. Image stacks were acquired using 250 nm Z-axis steps across a range of approximately 5 μm. Full stacks were recorded consecutively for each channel, from the lowest to the highest energy excitation wavelength. *Figure 6* data were recorded on an ONI Nanoimager S with an Olympus 100×super apochromatic oil immersion objective (NA 1.4). Dye excitation was performed with an ONI laser illumination system using the wavelengths 488 nm (EGFP) and 640 nm (Atto647N), with laser intensities set to 2% (488 nm) and 7% (641 nm). Fluorescent signals were recorded with a sCMOS camera with a pixel size of 0.117 μm. Images were acquired over a field of view of the camera chip resulting in a total imaging region of 50 μm×80 μm. Exposure times were adjusted accordingly to the signal intensity to avoid pixel saturation. Typical exposure times were 30ms for all channels. Images were recorded consecutively for each channel, from the lowest to the highest energy excitation wavelength. DNA-PAINT imaging was carried out on an inverted Nikon Eclipse Ti microscope (Nikon Instruments) equipped with the Perfect Focus System using objective-type total internal reflection fluorescence (TIRF) configuration (oil-immersion Apo SR TIRF, NA 1.49 100 x objective). A 200 mW 561 nm laser beam (Coherent Sapphire) was passed through a clean-up filter (ZET561/10, Chroma Technology) and coupled into the microscope objective using a beam splitter (ZT561rdc, Chroma Technology). Fluorescence light was spectrally filtered with an emission filter (ET575lp, Chroma Technology) and imaged with an sCMOS camera (Andor Zyla 4.2) without further magnification, resulting in an effective pixel size of 130 nm after 2×2 binning. Images were acquired using a region of interest of 512×512 pixels. The camera read-out rate was set to 540 MHz and images were acquired with an integration time of 200ms, using 50 W/cm$^2$ laser power. 5'-ATACATTGA-Cy3B-3' was used as imager strand sequence. Further details of imaging conditions for each experiment are summarised below (2 hpi sample: 1 nM imager, 20,000 frames; 4 hpi sample: 125 pM imager; 40,000 frames; 6 hpi: 100 pM imager; 30,000 frames).

## Image processing and colocalisation analysis

Deconvolution analysis was applied to all acquired widefield images using the Huygens Essential software (Scientific Volume Imaging B.V., the Netherlands). All channels and Z-planes were deconvolved using Huygens batch express tool (Standard profile).

Z-stacks were loaded with ImageJ and out-of-focus planes were manually discarded. Prior to analysis a maximum intensity Z-projection was performed with the remaining Z-planes.

2D colocalisation analysis was performed with maximum intensity Z-projections using Icy (Version 1.9.10.0), an open bioimage informatics platform (http://icy.bioimageanalysis.org/). Regions of interest were drawn around individual cells prior to the analysis. Pearson's correlation coefficient (PCC) was chosen as a statistic for quantifying colocalisation to measure the pixel-by-pixel covariance in the signal levels between two distinct channels. This allows subtraction of the mean intensity from each pixel's intensity value independently of signal levels and the signal offset for each ROI. Pearson's correlation coefficient values were calculated in Icy Colocalization Studio that employs pixel scrambling method *Lagache et al., 2015*.

For *Figure 1—figure supplement 1*, GAPDH and RV RNA transcripts peak intensities were detected using the ImageJ 'Find Maxima' function, with noise settings 100 (GAPDH) and 1000 (RV). Spot detections for intensity correlations (*Figure 2* and *Figure 2—figure supplement 1b*) were

performed using 'Spot Detection' tool in Icy. Spot detection input parameter sensitivity was set to 20, and the object size was set to 7 pixels. The sum intensity of each detection was calculated prior to the intensity correlation analysis. For spot intensities measured in *Figure 2—figure supplement 1c* and d, sensitivity was set to 20 and the object size to 3 pixels.

## DNA-PAINT image analysis

Raw fluorescence data were subjected to super-resolution reconstruction using Picasso software package *Jungmann et al., 2014*; *Schnitzbauer et al., 2017*. Drift correction was performed with a redundant cross-correlation and gold particles used as fiducial markers. The apparent on-rate (*apparent $k_{on}$*) of imager stands binding to their corresponding docking sites was used to quantify the relative number of binding sites. A higher *apparent $k_{on}$* value indicates a higher number of binding sites, i.e., RNA molecules detected in the structure *Jungmann et al., 2016*. $k_{on}$ values were calculated for each selected structure assuming $k_{on} = (\tau_D \times c)^{-1}$, where $c$ is the imager strand concentration and $\tau_D$ the dark time between binding events *Jungmann et al., 2016*. The number of FISH probes per RNA was calculated assuming $k_{on} = 1 \times 10^6 \, (Ms)^{-1}$ for each docking site. Further quantification and fitting were performed using OriginPro, as previously described *Jungmann et al., 2016*.

## Single-cell RNA imaging and viral transcriptome analysis using UDEx-FISH

Images were recorded with the same acquisition parameters for all rounds. Integrated signal densities and areas of single cells were measured in ImageJ to calculate signal intensities and areas for each RNA target. Background signals were determined from signals measured in mock-infected cells, and these were subtracted for each channel respectively. Signals in Cy3 and Cy5 channels were calibrated by calculating the correction factors for Cy3/Cy5 signals for one of the RNA targets that was imaged sequentially in both channels using Cy3 and Cy5 imager strands.

## Host cell and viral transcriptome RNA-Seq data analysis

RV-infected MA104 cells were harvested at 6 hpi, and total RNA was extracted using RNEasy kit (QIAGEN). One ug of total RNA was depleted of rRNA using NEBNext rRNA Depletion Kit (Human/Mouse/Rat) prior to cDNA synthesis primed with random hexanucleotide oligonucleotides (Random Primer 6, NEB). Sequencing library construction was carried out using NEBNext Ultra II FS DNA Library Prep Kit for Illumina (NEB), and the resulting library was sequenced using Illumina MiSeq v2 (2x150 bp) platform. Paired-end run raw sequencing data were pre-processed with fastp (v0.20.1) using default command line parameters *Chen et al., 2018* to yield 1,625,571 reads. A bwa-mem2 (v2.1) index was generated (default parameters) using bovine rotavirus A strain RF reference genome (J04346.1, KF729639.1, KF729643.1, KF729690.1, KF729653.1, K02254.1, KF729659.1, Z21640.1). Identified viral transcript reads had 99.81% identity against the reference genome. These preprocessed reads were mapped to the index using bwa-mem2 mem *Vasimuddin et al., 2019*. All mapped reads were sorted using samtools sort (v1.11) to create a consensus structure of all reads using bcftools (v1.11) *Li et al., 2009*; *Li, 2011*. bcftools mpileup (command line parameters: -Ob -d 10000) was used to generate genotype likelihoods, followed by bcftools call (command line parameters: -Ob -mv) for SNP and indel calling, bcftools norm (command line parameters: -Ob) for indel normalisation, bcftools index for indexing, and bcftools consensus was used to create the consensus structure. The consensus rotavirus RF genome file was combined with the transcriptome data available for *Chlorocebus sabaeus* sp. (MA104 host cell line) from Ensemble release 103 *Yates et al., 2020* to construct a combined transcriptome. This combined transcriptome file was used to generate a salmon (v1.4.0) index (command line parameters: `--keepDuplicates`) to quantify the pre-processed Illumina reads using salmon quant [6] (command line parameters: -l A `--validateMappings`) *Patro et al., 2017*. A total of 671,700 reads were identified by salmon (i.e., estimate of the number of reads mapping to each transcript that was quantified), out of which 114,276 reads were mapped to the viral transcriptome.

## Acknowledgements

We thank Dr Ulrich Desselberger (University of Cambridge, UK), Dr Elena Conti (Max Planck Institute of Biochemistry, Munich) for their valuable comments and suggestions during the preparation

of the manuscript. We also thank Dr Ulrich Desselberger (University of Cambridge, UK), Dr Oscar R Burrone (ICGEB, Trieste, Italy), and Dr Sarah McDonald Esstman (Wake Forest University, USA) for the generous gifts of the G6P6[1] strain RF, recombinant NSP3-2A-EGFP, and recombinant SA11 strain tsC rotavirus, respectively. The authors would like to thank Dr Martin Spitaler (MPIB, Munich) for technical assistance with imaging samples.

This work was supported by the Wellcome Trust (grants 103068/Z/13/Z and 213437/Z/18/Z to A.B.), ERC through an ERC Starting Grant (MolMap, Grant agreement number 680241), and the DGF through the SFB1032 (Nanoagents for the spatiotemporal control of molecular and cellular reactions, Project A11), the Max Planck Society, the Max Planck Foundation, and the Center for Nanoscience (CeNS). S.S. acknowledges support from the DFG through the Graduate School of Quantitative Biosciences Munich (QBM). This research was funded in part by the Wellcome Trust [213437/Z/18/Z]. For the purpose of Open Access, the author has applied a CC BY public copyright license to any Author Accepted Manuscript version arising from this submission.

## Additional information

### Funding

| Funder | Grant reference number | Author |
|---|---|---|
| Wellcome Trust | 213437/Z/18/Z | Alexander Borodavka |
| Deutsche Forschungsgemeinschaft | SFB1032 | Ralf Jungmann |
| European Research Council | MolMap 680241 | Ralf Jungmann |
| Max Planck Institute for Biochemistry | | Sebastian Strauss |

The funders had no role in study design, data collection and interpretation, or the decision to submit the work for publication. For the purpose of Open Access, the authors have applied a CC BY public copyright license to any Author Accepted Manuscript version arising from this submission. Open access funding provided by Max Planck Society.

### Author contributions

Sebastian Strauss, Resources, Data curation, Formal analysis, Investigation, Visualization, Writing – original draft; Julia Acker, Investigation, Visualization, Writing – review and editing; Guido Papa, Resources, Methodology, Writing – original draft; Daniel Desirò, Formal analysis, Methodology, Writing – original draft; Florian Schueder, Resources; Alexander Borodavka, Conceptualization, Resources, Software, Supervision, Funding acquisition, Investigation, Visualization, Methodology, Writing – original draft, Project administration, Writing – review and editing, Formal analysis, Data curation, Validation; Ralf Jungmann, Conceptualization, Resources, Data curation, Software, Funding acquisition, Visualization, Methodology, Writing – original draft

### Author ORCIDs

Guido Papa http://orcid.org/0000-0002-5215-0014
Florian Schueder http://orcid.org/0000-0003-3412-5066
Alexander Borodavka http://orcid.org/0000-0002-5729-2687
Ralf Jungmann http://orcid.org/0000-0003-4607-3312

### Decision letter and Author response

Decision letter https://doi.org/10.7554/eLife.68670.sa1
Author response https://doi.org/10.7554/eLife.68670.sa2

# Additional files

## Supplementary files
• Supplementary file 1. GenBank IDs and sequences of the DNA oligonucleotides used in the study.

• Supplementary file 2. RNA-Seq transcriptome analysis of RV-infected MA104 cells at 6 hpi (RVA strain RF).

• Transparent reporting form

## Data availability
RNA-Seq data have been uploaded, and the SRA Illumina reads data are available under the accession number PRJNA702157 (SRR13723918, RNA-Seq of Bovine Rotavirus A: Strain RF). SRA Metadata: BioProject: PRJNA702157 (Bovine rotavirus strain RF transcriptome of MA104 cells) BioSample: SAMN17926863 (Viral sample from Bovine rotavirus A) SRA: SRR13723918 (RNA-Seq of Bovine Rotavirus A: Strain RF) All data generated during this study are included in the manuscript and supporting files. Source data files have been provided for all figures. Primary image datasets (stacks of 3D DNA-PAINT files, widefield images) are available at: https://zenodo.org/record/5550075#.YXq6aXnTViN and https://zenodo.org/record/7470075#.Y6NblrLP05Q.

The following datasets were generated:

| Author(s) | Year | Dataset title | Dataset URL | Database and Identifier |
|---|---|---|---|---|
| Borodavka A | 2021 | Bovine rotavirus strain RF transcriptome of MA104 cells | https://www.ncbi.nlm.nih.gov/bioproject/PRJNA702157 | NCBI BioProject, PRJNA702157 |
| Strauss S, Acker J, Papa G, Desiró D, Schueder F, Borodavka A , Jungmann R | 2021 | Additional dataset for 'Principles of RNA recruitment to viral ribonucleoprotein condensates in a segmented dsRNA virus' | https://doi.org/10.5281/zenodo.7470075 | Zenodo, 10.5281/zenodo.7470075 |
| Strauss S, Borodavka A, Papa G, Acker J, Desiró D, Schueder F, Jungmann R | 2021 | Principles of RNA recruitment to viral ribonucleoprotein condensates in a segmented dsRNA virus | https://doi.org/10.5281/zenodo.5550075 | Zenodo, 10.5281/zenodo.5550075 |

The following previously published dataset was used:

| Author(s) | Year | Dataset title | Dataset URL | Database and Identifier |
|---|---|---|---|---|
| Yates A | 2020 | ENSEMBL 2020 | http://ftp.ensembl.org/pub/release-102/ | ensembl, 102 |

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
