## [Editor Report]

Viral replication in the cell requires the assembly of multiple viral components into individual viral particles, while maintaining a relatively strict ratio between individual components. This manuscript uses an imaging approach to study proposed aggregates of viral protein and nucleic acid components referred to as 'viroplasms' and their role in achieving ordered viral assembly. Although this work provides a glimpse of viral assembly in the cell, further work is required to better understand this complex and important process.

---

## [Decision Letter]

**Decision letter after peer review:**

Thank you for submitting your article "Principles of RNA recruitment to viral ribonucleoprotein condensates in a segmented dsRNA virus" for consideration by *eLife*. Your article has been reviewed by 2 peer reviewers, and the evaluation has been overseen by a Reviewing Editor and myself (Sara Sawyer – U Colorado Boulder – USA) as the Senior Editor. The following individuals involved in the review of your submission have agreed to reveal their identity: Roy Parker (Reviewer #1).

Essential revisions:

In this particular case, we are going to ask you to go through and address all reviewers' comments carefully. Both reviewers felt that the manuscript would be improved by careful quantification of all experiments and by putting the results into more rigorous analyses. We do feel that you will have a much stronger paper after addressing the comments of these reviewers, which will help you better interpret and explain your findings to readers.

*Reviewer #1 (Recommendations for the authors):*

1) In general, the manuscript would be improved by careful quantification of all experiments and putting the results into a more rigorous analysis.

2) I think the work would be much more interesting if additional analyses were performed by the images where all 11 segments are examined. For example:

a) Do all viroplasms have all 11 segments? If so, this implies specificity in the assembly mechanism that ensures this distribution.

b) Is the stoichiometry the same in each viroplasm, or is there significant variation between assemblies? Again, this would have implications for the assembly mechanisms.

I raise this issue since for publication in *ELife*, the work should provide new significant biological or mechanistic insight.

3) The manuscript would be improved by adding a co-localization quantification of the data presented in figure S5 (seg3 and seg4), which would communicate the importance of NSP2 more effectively than trying to gauge intensity from the images.

4) The analysis of the requirement for a specific 3' UTR is limited since there are multiple differences between the RNAs being examined. To really make this conclusion, one would need to express the positive control RNA from a nuclear transgene with a ribozyme to generate the proper 3' end, and show that nuclear RNA is still efficiently recruited to viroplasms. in addition, the mRNA with the poly(A) does seem to partition into the viroplasms to some extent. How does that affect the conclusions presented?

*Reviewer #2 (Recommendations for the authors):*

Strauss et al. studied the mechanism of RNA recruitment to ribonucleoprotein condenstates using rotavirus. They used multiplexed DNA-barcorded smFISH and DNA-PAINT for direct visualization of the RNP condensates in cells. They observed the early onset of viral transcript oligomerization before the formation of viroplasms and the process of enrichment in RNP condensates. They imaged all eleven transcript in a RNP condensate and quantified the amount of the transcripts. Based on these findings, they suggested a selective RNA enrichment mechanism of rotavirus. The authors conducted well the experiments with good control measurement. The results looks significant enough for understanding the RNA recruitment to RNP condensates and provide a potential usefulness of their approach in the future work. I have a few concerns before fully supporting its publication in *eLife*.

1. It may be very helpful for a non-expert in rotavirus to understand the purpose of this work by putting a schematic illustration of their biological systems in Supplementary Figures. For example, Figure 4 is easy to understand the experimental schemes and the results with the help of the illustration.

2. In the same manner, the conclusion of this work is rather difficult to understand. It may be very helpful to provide a schematic illustration of their results in a figure.

3. The last paragraph of introduction part. The authors provided all their results, which seems to make it difficult to understand the major findings of this work. It will be helpful to be concise here.

4. in Figure 1, the authors performed colocalization analysis. What is the error bar of their colocalization methods (how many pixels)? Please provide the dyes they used here for smFISH. Is there any cross-talk between EGFP and the dyes?

5. In Supplementary Figure 1. How did the authors obtained the intensity distribution of GAPDH? It is an intensity based on a pixel?

6. In Figure 3. The authors estimated 20-50 transcript for RNA clusters. When RNA formed clusters, the accessibility of DNA probes will be low compared with bare (or single) RNAs. The discussion on this point may be needed for the RNA quantification.

7. in page 13, "Assuming similar rates of transcription for each individual genomic segment, transcription of longer segments is expected to yield fewer copies of longer Seg1-Seg4 transcripts (3.4-2.6 kb)." The copy number of transcript typically depends on the promoter strength, i.e., initiation, not the length of transcript. Degradation rate may be different.

8. in Figure 5. It is difficult to compare the result in Figure 5c (RNA-seq) and 5d (FISH). Bar graphs (or other graphs) comparing the amount of transcripts for both methods will be helpful for understanding the results

[Editors’ note: further revisions were suggested prior to acceptance, as described below.]

Thank you for resubmitting the paper entitled "Principles of RNA recruitment to viral ribonucleoprotein condensates in a segmented dsRNA virus" for further consideration by *eLife*. Your revised article has been evaluated by a Senior Editor and a Reviewing Editor. We are sorry to say that we have decided that this submission will not be considered further for publication by *eLife*.

We have determined that several important reviewer comments from round 1 of the review have not been adequately addressed. In particular, any further consideration of this manuscript to *eLife* would need to address these points of Review #1 in a very solid and convincing way. The verbatim comments of Reviewer #1 are:

a) Do all viroplasms have all 11 segments? If so, this implies specificity in the assembly mechanism that ensures this distribution.

b) Is the stoichiometry the same in each viroplasm, or is there significant variation between assemblies? Again, this would have implications for the assembly mechanisms.

I raise this issue since for publication in *eLife*, the work should provide new significant biological or mechanistic insight.

[Editors’ note: further revisions were suggested prior to acceptance, as described below.]

Thank you for resubmitting your work entitled "Principles of RNA recruitment to viral ribonucleoprotein condensates in a segmented dsRNA virus" for further consideration by *eLife*. Your revised article has been evaluated by Miles Davenport (Senior Editor) and a Reviewing Editor.

The manuscript has been improved but there are some remaining issues that need to be addressed as outlined by reviewer 1 below.

*Reviewer #1 (Recommendations for the authors):*

This manuscript is significantly improved. Some comments to address before publication:

1) I think the manuscript would be strengthened by calculating the fraction of each RV segment in viroplasms at different time points. From looking at the images, it will probably be a small fraction and this would be of interest to the community going forward.

2) A formal caveat of the Nsp2 shRNA experiment is that the Nsp2 encoding RNA could be required for the viroplasm assembly. I think the authors should explicitly acknowledge this limitation.

3) The manuscript would be greatly strengthened by quantification of Figure 6 and 7, which otherwise just show one image (and the GFP intensity in the images in Figure 6 looks strikingly different, which could affect the result). This would be important to make this experiment robust.

---

## [Author Response]

Reviewer #1 (Recommendations for the authors):1) In general, the manuscript would be improved by careful quantification of all experiments and putting the results into a more rigorous analysis.2) I think the work would be much more interesting if additional analyses were performed by the images where all 11 segments are examined. For example:a) Do all viroplasms have all 11 segments? If so, this implies specificity in the assembly mechanism that ensures this distribution.b) Is the stoichiometry the same in each viroplasm, or is there significant variation between assemblies? Again, this would have implications for the assembly mechanisms.I raise this issue since for publication in ELife, the work should provide new significant biological or mechanistic insight.

We thank the Reviewer for these comments. We carried out additional analyses of the imaging data, and we have provided further comments and interpretation throughout the text. More specifically, we have shown that all analysed viroplasms contained 11 distinct RNA transcripts, albeit the transcript stoichiometry appears to deviate from the equimolar ratio of dsRNA segments present in each rotavirus particle (Figure 5). However, we cannot rule out that this would universally apply to all viroplasms, as we have only analysed those detectable around 6 hours post infection (HPI), when all transcripts are highly abundant in RV-infected cells. Furthermore, since viroplasms represent RNP granules with poorly defined RNA and protein stoichiometries, we expect significant variation in the RNA content between individual viroplasms. To test this, we have carried out additional quantitative analyses of Seg3 and Seg4 RNAs in viroplasms detected at 4 HPI and 6 HPI (Figure 2c and Figure 2 —figure supplement 3, b). Figure 2c reveals a strong correlation (R^2^ = 0.93) between Seg4 and Seg3 RNA intensities (overall intensities per detected spot per RNA target), both of which linearly increase as the intensity of the NSP5-EGFP signal increases, i.e., proportionally to the size of an EGFP-NSP5-tagged viroplasm. Interestingly, a similar Seg3/Seg4 RNA ratio is also maintained in viroplasms at 6 HPI (R^2^=0.9), however, the RNA:EGFP-NSP5 ratio is distinct from that seen at 4 HPI (R^2^=0.8 for 4 HPI and R^2^ = 0.13 for 6 HPI). These observations suggest that while all 11 distinct transcripts are present in viroplasms, transcript : protein ratios are likely to change during infection, reflecting the dynamic composition of viroplasmic RNP granules.

3) The manuscript would be improved by adding a co-localization quantification of the data presented in figure S5 (seg3 and seg4), which would communicate the importance of NSP2 more effectively than trying to gauge intensity from the images.

We thank the Reviewer for their suggestion, and we have significantly improved the layout of Figure 2 —figure supplement 4 by (i) adding schematic diagrams of the experimental system; (ii) adding signal intensity analysis for both Seg3 and Seg4 for RV-infected WT MA104 cells, and after shRNA-mediated depletion of NSP2, as suggested by the Reviewer. As you can see from the signal intensity analysis, the overall RNA signal intensities in shRNA-expressing cells were much lower than those observed in WT cells. Thus, we refrained from carrying out colocalization analysis due to significant differences in signal intensities to avoid biases introduced by significantly different thresholds required for estimating colocalizations between these conditions (i.e., in images with low signal levels an appropriate threshold value is much lower and it fails to discriminate RNA signals from background (Dunn KW, Kamocka MM & McDonald JH. A practical guide to evaluating colocalization in biological microscopy. Am J Physiol Cell Physiol, 2011)). More importantly, high intensity foci containing colocalising Seg3 and Seg4 RNAs were only present in RV-infected WT cells, whereas no RNA clusters containing both Seg3 and Seg4 RNAs could be detected in NSP2-depleted cells after RV infection.

4) The analysis of the requirement for a specific 3' UTR is limited since there are multiple differences between the RNAs being examined. To really make this conclusion, one would need to express the positive control RNA from a nuclear transgene with a ribozyme to generate the proper 3' end, and show that nuclear RNA is still efficiently recruited to viroplasms. in addition, the mRNA with the poly(A) does seem to partition into the viroplasms to some extent. How does that affect the conclusions presented?

We thank the Reviewer for their comment above, we agree that the presented data do not rule out the possibility of other factors being involved in viral transcript partitioning into viroplasmic condensates. However, expressing viral transcript as a transgene would pose many additional challenges. Specifically, such a strategy could be confounded by (i) cryptic splice sites resulting in splicing of the exported viral transcript; (ii) processing of the 3’ end during nuclear export; (iii) processing of the 3’ end by the HDV ribozyme in the nucleus, resulting in the loss of its nuclear export. We expanded our discussion to mention that association of cellular RBPs with the viral transcripts exported from the nucleus may also impede their partitioning into viroplasms. We have also mentioned that given the high density of NSP5-EGFP in viroplasms, it is hard to rule out the possibility of residual cross-talk between high-density EGFP and the low-level RNA FISH signal (Atto647N-labelled FISH probes, please see the spectral data in Author response image 1, EGFP Excitation/Emission in green, Atto647N Excitation/Emission spectra in red).

**Author response image 1. sa2fig1:** 

Nevertheless, our experimental data show that non-viral, heterologous EGFP sequences inserted into the RV transcripts containing intact UTRs would partition into viroplasms much more efficiently compared to nuclear transcripts. These data, together with our observation that GAPDH transcripts are also not enriched in viroplasms, suggest that poly-adenylated transcripts do not efficiently partition into viroplasmic condensates.

Reviewer #2 (Recommendations for the authors):Strauss et al. studied the mechanism of RNA recruitment to ribonucleoprotein condenstates using rotavirus. They used multiplexed DNA-barcorded smFISH and DNA-PAINT for direct visualization of the RNP condensates in cells. They observed the early onset of viral transcript oligomerization before the formation of viroplasms and the process of enrichment in RNP condensates. They imaged all eleven transcript in a RNP condensate and quantified the amount of the transcripts. Based on these findings, they suggested a selective RNA enrichment mechanism of rotavirus. The authors conducted well the experiments with good control measurement. The results looks significant enough for understanding the RNA recruitment to RNP condensates and provide a potential usefulness of their approach in the future work. I have a few concerns before fully supporting its publication in eLife.

We thank the Reviewer for their comments, and for appreciating the usefulness of our approach for investigating RNA composition of RNP granules.

1. It may be very helpful for a non-expert in rotavirus to understand the purpose of this work by putting a schematic illustration of their biological systems in Supplementary Figures. For example, Figure 4 is easy to understand the experimental schemes and the results with the help of the illustration.

We thank the Reviewer for this comment – we have added new schematic diagrams (specifically, Figure 1, Figure 2, Figure 2 —figure supplement 4, Figure 3) to clarify the experimental designs, as well as a new Figure 7 to summarize our findings. We hope the readers will find these new schematics helpful and improve the overall narrative of the manuscript.

2. In the same manner, the conclusion of this work is rather difficult to understand. It may be very helpful to provide a schematic illustration of their results in a figure.3. The last paragraph of introduction part. The authors provided all their results, which seems to make it difficult to understand the major findings of this work. It will be helpful to be concise here.

We thank the Reviewer for their excellent suggestion – we have re-written the Introduction section to highlight the major findings in the context of viroplasm formation and RNP selectivity towards RNAs.

4. in Figure 1, the authors performed colocalization analysis. What is the error bar of their colocalization methods (how many pixels)? Please provide the dyes they used here for smFISH. Is there any cross-talk between EGFP and the dyes?

The errors are included in the violin plots shown in panel (e). Each data point represents the PCC value calculated for a single cell, i.e., the whole image was used for calculating PCCs. We have added the information about the fluorophores (Quasar570-labelled FISH probes) in the Figure legend. All images were acquired by sequential excitation with the lowest energy to the highest energy laser lines to minimise potential cross-talk between EGFP and Quasar570 dye, as mentioned in the Materials and methods section.

5. In Supplementary Figure 1. How did the authors obtained the intensity distribution of GAPDH? It is an intensity based on a pixel?

We have added the following information in the Materials and methods section:

‘For Figure 1 —figure supplement 1, GAPDH and RV RNA transcripts peak intensities were detected using the ImageJ ‘Find Maxima’ function, with noise settings 100 (GAPDH) and 1000 (RV). Spot detections for intensity correlations (Figure 2 and Figure 2 —figure supplement 3) were performed using ‘Spot Detection’ tool in Icy (Version 1.9.10.0), an open bioimage informatics platform (http://icy.bioimageanalysis.org/). The spot detection input parameter sensitivity was set to 20 and the object size to 7 pixels. The sum intensity of each detection was calculated prior to the intensity correlation analysis. For spot intensities measured in Figure 2 —figure supplement 4 sensitivity was set to 20 and the object size to 3 pixels.’

6. In Figure 3. The authors estimated 20-50 transcript for RNA clusters. When RNA formed clusters, the accessibility of DNA probes will be low compared with bare (or single) RNAs. The discussion on this point may be needed for the RNA quantification.

We thank the Reviewer for making this excellent point. Indeed, target accessibility inside the dense phase of RNP granules/other phase-separated systems remains one of many challenges for quantitative imaging analyses of their composition. We have added the following sentence in the Results section: ‘We note that within viroplasms, target accessibility for smFISH probes is expected to be lower than that for single transcripts. Thus, Seg3 transcripts present in these RNA clusters and larger viroplasms are likely to be undercounted’.

7. in page 13, "Assuming similar rates of transcription for each individual genomic segment, transcription of longer segments is expected to yield fewer copies of longer Seg1-Seg4 transcripts (3.4-2.6 kb)." The copy number of transcript typically depends on the promoter strength, i.e., initiation, not the length of transcript. Degradation rate may be different.

We have modified this statement to highlight that in rotaviruses, the RdRP-binding sites for each individual dsRNA segment are identical, and their segment-specific transcription rates deduced from previous in vitro studies suggest that the number of transcripts produced by a transcriptionally active particle is inversely proportional to their length (e.g., shorter transcripts are more abundant than the longer transcripts). We have added a sentence ‘Indeed, purified transcriptionally active DLPs yield similar amounts of individual transcripts produced in vitro’.

8. in Figure 5. It is difficult to compare the result in Figure 5c (RNA-seq) and 5d (FISH). Bar graphs (or other graphs) comparing the amount of transcripts for both methods will be helpful for understanding the results

Thank you for this comment, we have modified the figure, and we also added a Figure 5 —figure supplement 1.

[Editors’ note: further revisions were suggested prior to acceptance, as described below.]

We have determined that several important reviewer comments from round 1 of the review have not been adequately addressed. In particular, any further consideration of this manuscript to eLife would need to address these points of Review #1 in a very solid and convincing way. The verbatim comments of Reviewer #1 are:a) Do all viroplasms have all 11 segments? If so, this implies specificity in the assembly mechanism that ensures this distribution.b) Is the stoichiometry the same in each viroplasm, or is there significant variation between assemblies? Again, this would have implications for the assembly mechanisms.I raise this issue since for publication in eLife, the work should provide new significant biological or mechanistic insight.

We thank the Reviewer for these comments. We carried out additional analyses of the imaging data, and we have provided further comments and interpretation throughout the text. Specifically, we have shown that all viroplasms contained 11 distinct RNA transcripts, however, the transcript stoichiometry varies between individual viroplasms, particularly for shorter segments. These results are now included in Figure 5b and Figure 5—figure supplement 1.

[Editors’ note: further revisions were suggested prior to acceptance, as described below.]

The manuscript has been improved but there are some remaining issues that need to be addressed as outlined by reviewer 1 below.Reviewer #1 (Recommendations for the authors):This manuscript is significantly improved. Some comments to address before publication:1) I think the manuscript would be strengthened by calculating the fraction of each RV segment in viroplasms at different time points. From looking at the images, it will probably be a small fraction and this would be of interest to the community going forward.

As we have shown in the revised version of the manuscript, even at the same infection time point within the same cell, the number of transcripts greatly varies across viroplasms. This can be seen from Figure 2 (quantitation of Seg3 and Seg4 RNAs). Further quantitation of the RV transcriptome in viroplasm at different time points across multiple cells would require a significant investment in time and resources and would extend beyond the scope of the original manuscript.

2) A formal caveat of the Nsp2 shRNA experiment is that the Nsp2 encoding RNA could be required for the viroplasm assembly. I think the authors should explicitly acknowledge this limitation.

Excellent point – we have added the following statement in Results section ‘Although we observed a lack of transcript clustering in rotavirus-infected cells expressing shRNA targeting NSP2, it is important to note that the formation of viroplasms was also impaired under these conditions, supporting the essential role of NSP2 in viroplasm assembly’ (p.5). We have also added the following statement in Discussion: ‘While NSP2 has been demonstrated to promote RNA oligomerization in vitro, the shRNA-mediated knockout of NSP2 may also disrupt RNA distribution and localization due to the impaired viroplasm formation, suggesting a potential role for NSP2 in these processes.’ (p.8).

3) The manuscript would be greatly strengthened by quantification of Figure 6 and 7, which otherwise just show one image (and the GFP intensity in the images in Figure 6 looks strikingly different, which could affect the result). This would be important to make this experiment robust.

We thank the reviewer for their comment regarding Figure 6. We inspected all datasets carefully, and we agree that the GFP background appears to be higher, also due to the lower intensity of the transfected RNA, for which we had to use longer exposure times. In addition, we had our concerns that covalent ATTO647N dye-labelling of the RNA could also impact the physicochemical properties of labelled transcripts (e.g., addition of multiple cationic and hydrophobic ATTO647N dye molecules). We therefore devised a modified strategy that avoids the use of dye labelled RNAs and relies on the following:

(i) Production of unmodified, capped EGFP mRNA containing intact UTRs derived from Segment 1 RNA. We chose the EGFP ORF sequence as a target since we have already validated our EGFP-targeting FISH probes labelled with the 647 dye.

(ii) NSP2-mCherry cell line was then electroporated with unmodified capped EGFP mRNAs containing either both 5’ and 3’ UTRs derived from Segment 1 RNA, or without the 3’UTR (denoted EGFP-D3’UTR), and then these cells were infected with rotaviruses, subsequently fixed and EGFP mRNA distribution was analysed by smFISH.

(iii) Imaging viroplasms using the NSP2-mCherry MA104 cell line was carried out as both NSP5-EGFP and NSP2-mCherry are equally susceptible to RVs, both being useful for visualising viroplasms (doi.org/10.15252/embj.2021107711). mCherry has approximately two-fold lower molecular brightness compared to EGFP (15,840 vs 33,540, respectively, based on their extinction coefficients and the quantum yields), while its absorption spectrum does not overlap with the 641 nm excitation line required for smFISH visualisation of transfected EGFP mRNAs. This allowed us to further reduce the fluorescence background, while EGFP-specific probes do not cross-react with mCherry transcripts (53.6% sequence similarity between mCherry and EGFP).

The new images further support the notion that functional, UTR-containing non-viral EGFP mRNAs electroporated to MA104 cells would partially localise to viroplasms in RV-infected cells. We have included these new data in Figure 6, and we updated Methods section.

Furthermore, for Figure 7, we have also included quantitation of colocalizing signals, as requested by the Reviewer. The Pearson’s correlation coefficients (PCC) and associated errors are now included in the revised manuscript (Figure 7b, R^2^ = 0.78; Figure 7c, R^2^ = 0.62 vs R^2^ = 0.18; please see revised figure legends, and Results section, highlighted in red, pp. 8)